# Positive Indian Ocean Dipole events prevent anoxia off the west coast of India

V. Parvathi[1], I. Suresh[1], M. Lengaigne[2,3], C. Ethé[2], J. Vialard[2], M. Levy[2], S. Neetu[1], O. Aumont[2], L. Resplandy[4], H. Naik[1], S. W. A. Naqvi[1]

[1]CSIR-National Institute of Oceanography, Dona Paula, Goa 403 004, India
[2]LOCEAN-IPSL, Sorbonne Universités, LOCEAN (UPMC/CNRS/IRD/MNHN), Paris, France
[3] Indo-French Cell for Water Sciences, IISc-NIO-IITM–IRD Joint International Laboratory, NIO, Goa, India
[4]Scripps Institution of Oceanography, La Jolla, California, United States

*Correspondence to*: I. Suresh (isuresh@nio.org)

**Abstract.** The seasonal upwelling along the west coast of India (WCI) brings nutrient-rich, oxygen-poor subsurface waters to the continental shelf, favoring very low oxygen concentrations in the surface waters during late boreal summer and fall. This yearly-recurring coastal hypoxia is more severe during some years, leading to coastal anoxia that has strong impacts on the living resources. In the present study, we analyze a ¼°-resolution coupled physical-biogeochemical regional oceanic simulation over the 1960-2012 period to investigate the physical processes influencing the oxycline interannual variability

off the WCI, that being a proxy for the variability on the shelf in our model. Our analysis indicates a tight relationship between the oxycline and thermocline variations in this region at both seasonal and interannual timescales, thereby revealing a strong physical control of the oxycline variability. As in observations, our model exhibits a shallow oxycline/thermocline during fall that combines with interannual variations to create a window of opportunity for coastal anoxic events. We further demonstrate that the boreal fall oxycline fluctuations off the WCI are strongly related to the Indian Ocean Dipole (IOD),

with an asymmetric influence of its positive and negative phases. Positive IODs are associated with easterly wind anomalies near the southern tip of India. These winds force downwelling coastal Kelvin waves that propagate along the WCI and deepen the thermocline and oxycline there, thus preventing the occurrence of coastal anoxia. On the other hand, negative IODs are associated with WCI thermocline and oxycline anomalies of opposite sign, but of smaller amplitude, so that the negative or neutral IOD phases are necessary, but not the sufficient condition for coastal anoxia. As the IODs generally start

developing in summer, these findings suggest some predictability to the occurrence of coastal anoxia off the WCI a couple of months ahead.

## 1 Introduction

The continental shelf off the west coast of India (WCI) is home to the largest coastal hypoxic system of the world ocean, spreading over an area of ~180,000 km$^2$ (Naqvi et al., 2000). These hypoxic conditions, characterized by oxygen concentration lower than 20 μmol.l$^{-1}$, occur in fall, right after the southwest monsoon. Importantly, substantial year-to-year changes in both the duration and intensity of this seasonal oxygen deficiency have been reported in the literature (e.g. Naqvi et al., 2009). While the oxygen concentrations in near-bottom waters are systematically low enough to trigger conversion of oxidized nitrogen to molecular nitrogen, mostly through denitrification, this deficiency in some years is even more severe and the bottom waters turn sulphidic, a condition called coastal anoxia (Naqvi et al., 2006). These anoxic events have tremendous impact on living resources (e.g. Diaz and Rosenberg, 2008), with more frequent episodes of fish mortality and a shorter span of fishing season, inducing a sharp decline in fish catches.

Observations from a series of ship cruises during September – October 1999 (Naqvi et al., 2000) off the WCI and time-series measurements from a fixed site off Goa since 1997 (Naqvi et al., 2009) indicate the occurrence of severe hypoxic conditions over almost the entire shelf and anoxic conditions close to the WCI, with the most intense anoxic event reported in fall 2001 and moderate ones in fall 1998 and 1999. In contrast, these data indicate that fall 1997 was characterized by far less hypoxic conditions. The frequent anoxic conditions occurred during the 1998-2002 period were accompanied by a three to five-fold decline in demersal fish catch in 1999 and 2001 compared to 1997. The total fish landing also remained low between 1998 and 2002, adversely affecting the economy from fisheries, and the pelagic fish catches shifted from the dominance of mackerel to oil sardine from 1998 to 1999 (Krishnakumar and Bhat, 2008). There has been a remarkable revival of fisheries since 2003, apparently due to a relaxation of the intensity of oxygen deficiency, with no severe anoxic event reported over the recent years. Subsurface oxygen concentrations have also been reported to be significantly lower for the 1997–2004 period than for the 1971–1975 period (Naqvi et al., 2009). These observations thus reveal large interannual and decadal fluctuations in the oxygen deficiency along the WCI, but the processes responsible for these variations have not yet been understood (Naqvi et al., 2009).

As opposed to the other coastal hypoxic systems that have generally developed as a result of human activities (largely eutrophication) in the last few decades (Diaz and Rosenberg, 2008), the seasonal surface oxygen deficiency along the WCI is naturally driven. Indeed, it has been suggested that the oxygen-deficient conditions that develop in early fall along the WCI result from the seasonal upwelling, which brings poorly oxygenated sub-surface waters from the Oxygen Minimum Zone (OMZ) in the interior Arabian Sea (e.g. Morrison et al., 1999; Naqvi, 1987; Sarma, 2002) towards the surface along the continental shelf. This connection between the offshore and the shelf oxygen content has been discussed by several studies on the basis of in-situ oxygen observations along different cross-shelf transects. For example, Banse (1959) showed a clear

association between the seasonal upwelling and coastal hypoxia on the shelf off Cochin at ~10°N in the last week of August 1957. A similar connection has also been reported by Carruthers et al. (1959) off Mumbai at 19°N in early November 1958 and later by Naqvi et al. (2009) off Mangalore at 13°N by late summer for several years. A recent study by Gupta et al. (2016) used ten shelf transects near $10^0$ N during 2012 to conclude that the upwelling of oxygen-deficient waters along the shelf break during the monsoon is the major process regulating the biogeochemistry on the shelf. Although these studies indicate a clear connection between the shelf oxygen content and upwelling at the shelf break, the actual processes in these exchanges are not well understood, owing to the lack of continuous and high frequency observations.

The seasonal upwelling along the WCI starts in April but the shallowest thermocline marking the peak of the upwelling is usually observed during September-October (e.g. Schott and McCreary, 2001). Local alongshore winds are however only favorable to upwelling during the southwest monsoon (i.e., they only have an equatorward alongshore component from June to August). This is an indication that upwelling along the WCI is to a large extent forced by remote winds (Shetye et al., 1990). Wind variations in the equatorial band and Bay of Bengal indeed force coastal Kelvin waves that travel along the rim of the bay and up the WCI to influence the thermocline depth there (e.g. McCreary et al. 1993; Shetye, 1998). Recently, Suresh et al. (2016) demonstrated that wind variations in the vicinity of Sri Lanka are responsible for a large fraction of the seasonal upwelling along the WCI. On contrary to this seasonal variability of the upwelling and upper-ocean oxygen, there are currently little clues as to what causes the interannual variability along the WCI.

Identifying the main drivers of the near-surface oxygen interannual variations in regions of the main coastal hypoxic systems is an important endeavor as it may ultimately help to predict the occurrence of severe anoxic events. The large-scale climate modes have been suggested to influence the local oxygen variability in several coastal regions. For instance, the El Niño-Southern Oscillation (ENSO) strongly influences the oxygen concentrations along the coasts of Peru and Chile (Helly and Levine, 2004; Arntz et al., 2006; Gutierrez et al., 2008), with intensified oxygenation associated with weak El Niño upwelling and intensified hypoxia associated with strong La Niña upwelling. As in the Pacific, the natural climate variability in the Indian Ocean could also be a potential candidate responsible for the near-surface oxygen interannual variations along the WCI, but that has so far not been evaluated. The main indigenous mode of Indian Ocean interannual climate variability is the Indian Ocean Dipole (hereafter IOD; Saji et al., 1999; Webster et al., 1999; Murtugudde et al., 2000). A "positive" IOD is characterized by a cooling and anomalously shallow thermocline in the eastern Indian Ocean, and by a warming and anomalously deep thermocline in the central and western Indian Ocean, driven by anomalous easterlies in the central equatorial Indian Ocean. A "negative" IOD is associated with anomalous signals of opposite polarities. The IOD usually starts developing during boreal summer and peaks in fall (e.g. Saji et al., 1999). El Niño and La Niña events tend to induce respectively the positive and negative IODs in the Indian Ocean, but IOD can also occur independent of ENSO (e.g. Annamalai et al., 2003). The IODs induce larger-amplitude, large-scale wind and thermocline-depth variations than those

associated with El Niño events over the Indian Ocean (e.g. Currie et al., 2013) and thus have the potential to affect the upwelling variations along the WCI through planetary wave propagation.

While the spatio-temporal density of observations in the eastern Pacific margin has allowed to accurately describe the monthly oxygen evolution along the west coast of South America over the past three decades (Helly and Levine, 2004; Arntz
et al., 2006), there is a dearth of long-term data from fixed sites in the Indian Ocean as compared to the Pacific and the Atlantic (Gupta et al., 2016), which in turn prevents building a reliable time series that could depict the year-to-year variations. On the other hand, three-dimensional coupled physical-biogeochemical models that include the oxygen cycle have contributed to improve the description and understanding of dynamics of hypoxic events in various coastal regions (Pena et al., 2010), such as the Gulf of Mexico (e.g. Hetland and DiMarco, 2008), Black Sea (e.g. Gregoire and Friedrich,
2004) and Baltic Sea (e.g. Eilola et al., 2009). Such models have also been used to investigate the mechanisms driving the spatial distribution (McCreary et al., 2013) and seasonal evolution of the OMZ in the interior Arabian Sea (Resplandy et al., 2012). There is, however, no dedicated modeling study to date, addressing the mechanisms that drive the interannual oxygen variability along the WCI. The present study aims at identifying the physical controls of the WCI interannual oxygen variability, with the help of a 40-year long simulation from a ¼° regional coupled physical-biogeochemical model. While the
~25-km resolution of our model is insufficient to describe all the physical processes that influence the coastal upwelling (e.g. Huthnance 1995; Allen et al. 2010) or the shelf-open ocean interactions, we will show that our model still reproduces the essential characteristics of the large scale oxygen fluctuations off the shelf break, and that the offshore fluctuations provide a proxy for the variations on the shelf (Banse, 1959; Carruthers et al., 1959). Section 2 describes our model, data and methods, and provides a brief evaluation of the model. Section 3 describes the main seasonal features of the thermocline and oxycline
variability along the WCI in both model and observations. The strong influence of the IOD on the interannual oxycline variations along the WCI is then assessed from the model analysis in section 4. Section 5 summarizes our results, discusses them in the context of earlier studies and assesses the limitations of our approach.

**2 Data and methods**

**2.1 Observations**

We used the 1°-resolution World Ocean Atlas-2013 (WOA13; Boyer et al., 2013) for evaluating the model accuracy at representing the large-scale climatological temperature and oxygen. The sea level is a good proxy for vertical movements of the thermocline in tropical regions (e.g. Fukumori et al., 1998), and hence we used the sea-level data from satellite altimetry produced by Ssalto/Duacs and distributed by AVISO (http://www.aviso.altimetry.fr/duacs/) as a proxy for the thermocline interannual variability along the WCI. In addition, we used the monthly level 3 Ocean Color Climate Change Initiative (OC-
CCI) product, available at http://www.oceancolour.org/, that merges data from the SeaWiFs, MERIS and MODIS ocean color missions to evaluate the model surface chlorophyll climatological seasonal cycle.

We also used oxygen measurements from Candolim Time Series (CaTS) station located on the WCI in the inner shelf off Goa (~15.5°N; 73.6°E) to construct a seasonal cycle of oxygen variations near the coast. This station has been established by Council of Scientific and Industrial Research-National Institute of Oceanography (CSIR-NIO) and records the physical (temperature, salinity) and biogeochemical (oxygen, nitrate, nitrite, hydrogen sulphide, etc.) parameters from September

1997 onwards (see Maya et al., 2011 for a detailed description). This site lies approximately 10 km off the Candolim beach, where the depth of the water column is ~28m. Samples are taken at four depths: just below the surface, just above the bottom, and at two intermediate depths equally spaced between the surface and bottom. This dataset consists of a total of 142 vertical profiles during the 1997-2010 period and has been extensively used by the previous studies (e.g., Naqvi et al., 2006, 2009, 2010a). Fig. 1a shows the percentage of years sampled month-wise in this dataset. It indicates that in-situ

measurements have been performed at this station ~50% of the years over the 1997-2010 period for most calendar months, including September-October (the time of peaking of anoxic events), except for June (< 10%) and July (no data), when summer monsoon rough weather conditions prevent observations (Naqvi et al., 2009). This temporal coverage allows us to build a reliable monthly climatology of near-coastal oxygen variations, except for the month of June and July. Though CaTS offers the best available dataset to study the oxygen variations along the WCI, a thorough description of oxygen interannual

variability is limited by two reasons. First, during our period of interest, i.e., September – October, the temporal distribution of the data indicates that only ~50% of the years were sampled over the 1997-2010 period. Second, the months for which a large number of records are available reveal very different vertical oxygen profile over the same month (like in September 1998 or 2000; Figs. 1c and 1e), highlighting the existence of a large variability at sub-monthly timescale. For some other months, the number of available profiles is very limited, like in September 1999, when only one profile is available (Fig. 1d).

Given the existing high frequency variability, averaging this very limited number of profiles may not provide a value representative of the actual monthly average. As a result, the uneven temporal distribution and the sub-monthly variability do not allow us to build a reliable monthly pluri-annual time series from this dataset. In addition, these observations collected at a single coastal site may be influenced by local processes and hence may not be representative of the dynamics along the entire WCI. We will hence identify the years when severe oxygen deficiency occurs from Table 1 of Naqvi et al. (2009),

which is based on the absence of nitrate and nitrite and the presence of hydrogen sulphide in the water column. These years will be further discussed and compared with our model results at the end of this paper.

## 2.2 Model description

This study uses the NEMO (Nucleus for European Modelling of the Ocean; Madec, 2008) model that includes the PISCES (Pelagic Interaction Scheme for Carbon and Ecosystem Studies; Aumont et al., 2015) biogeochemical component. The

PISCES model has 24 compartments, which include two sizes of sinking particles and four "living" biological pools, representing two phytoplankton (nano-phytoplankton and diatoms) and two zooplankton (microzooplankton and meso-zooplankton) size classes. Phytoplankton growth is limited by five nutrients: NO3, NH4, PO4, SiO4, and Fe. The ratios

---

**Matthieu Lengaigne 6/3/2017 1:16 PM**
**Deleted:** The model used in

**Matthieu Lengaigne 6/3/2017 1:16 PM**
**Deleted:** t

**Matthieu Lengaigne 6/3/2017 1:16 PM**
**Deleted:** couples

**Matthieu Lengaigne 6/3/2017 1:17 PM**
**Deleted:** physical ocean component

**Matthieu Lengaigne 6/3/2017 1:17 PM**
**Deleted:** with

**Parvathi V 2/3/2017 10:26 AM**
**Comment [1]:** http://pcwww.liv.ac.uk/~atagliab/LIV_WEB/Global_Models.html

**Matthieu Lengaigne 6/3/2017 1:14 PM**
**Deleted:** through the OASIS3 (Valcke, 2013) coupler

**Matthieu Lengaigne 6/3/2017 1:14 PM**
**Deleted:** In the present study we use an OGCBM (Ocean General Circulation and biogeochemistry Model), NEMO (Nucleus for European Modelling of the Ocean; Madec, 2008). The biogeochemical component of NEMO is PISCES (Pelagic Interaction Scheme for Carbon and Ecosystem Studies; Aumont et al., 2015).

among C, N, and P are kept constant for the "living" compartments, at values proposed by Takahashi et al. (1985). The internal Fe contents of both phytoplankton groups and Si contents of diatoms are prognostically simulated as a function of ambient concentrations in nutrients and light level. Details on the red-green-blue model from which light penetration profiles are calculated are given in Lengaigne et al. (2007). The Chl/C ratio is modeled using a modified version of the photo-adaptation model by Geider et al. (1998). Dissolved oxygen is prognostic and evolves in response to physical conditions (advection, mixing), biological sources and sinks and the air-sea fluxes:

$$\partial_t O_2 = \underbrace{\left(\frac{\partial O_2}{\partial t}\right)_{Dyn}}_{Dynamical\,Transport} + \underbrace{\left(\frac{\partial O_2}{\partial t}\right)_{Bio}}_{Biological\,sources\ and\,sinks} + J_{flux}$$

where $(\partial O_2/\partial t)_{Bio}$ includes all biological processes affecting the concentration of $O_2$, $(\partial O_2/\partial t)_{Dyn}$ accounts for large scale and turbulent transport of oxygen and $J_{flux}$ is the contribution of $O_2$ air–sea fluxes. The response of oxygen to biological processes $(\partial O_2/\partial t)_{Bio}$ is computed as follows:

$$\left(\frac{\partial O_2}{\partial t}\right)_{Bio} = \underbrace{\left(R_{o:c}^1 + R_{o:c}^2\right)\left(\mu_{NO_3}^P P + \mu_{NO_3}^D D\right)}_{New\ Production}$$
$$+ \underbrace{R_{o:c}^1\left(\mu_{NH_4}^P P + \mu_{NH_4}^D D\right)}_{Regenerated\,Production} - \underbrace{\lambda_{DOC}^* f(O_2) DOC}_{Remineralization}$$
$$\underbrace{-G^Z Z - G^M M}_{Respiration} - \underbrace{R_{o:c}^2 Nitrif}_{Nitrification}$$

Oxygen is produced during net primary production (calculating the uptake of nitrate and ammonium by phytoplankton separately) by nanophytoplankton (P) and diatoms (D) and consumed by dissolved organic matter (DOC) remineralization, small (Z) and large (M) zooplankton respiration, and nitrification. This last term represents the conversion of ammonium into nitrate and is assumed to be photo-inhibited, and reduced in suboxic waters. It is therefore a function of the ammonium and oxygen concentrations and photosynthetically available radiation. In this equation, the stoichiometric ratio $R_{o:c}^1$ represents the change in oxygen relative to carbon during ammonium conversion into organic matter, whereas $R_{o:c}^2$ denotes the consumption of oxygen during nitrification. Their values have been set respectively to 131:122 and 32:122 so that the typical Redfield ratio for oxygen is equal to 1.34 as proposed by Kortzinger et al. (2001).

When oxygen falls below a threshold value set to 6 $\mu$mol.L$^{-1}$, nitrate instead of oxygen starts to be increasingly consumed during the remineralization of organic matter, i.e., denitrification. To avoid negative oxygen concentration in the model, all processes consuming oxygen are switched off below a 10$^{-3}$ $\mu$mol.L$^{-1}$ threshold. Anammox is not represented in the model. At the bottom of the ocean, the model includes a very simple description of the sediment processes. The metamodel

of Middelburg et al. (1996) is further used to compute the relative contribution of denitrification to the remineralization of the organic matter. Then, the flux of organic matter to the sediment is used to compute the oxygen and nitrate demands in the sediment, which are then imposed as boundary conditions to the model.

5 The response of oxygen to dynamical processes is computed as:

$$\left(\frac{\partial O_2}{\partial t}\right)_{Dyn} = \underbrace{-u_H \cdot \nabla_H O_2}_{lateral\ advection} - \underbrace{w \cdot \nabla_H O_2}_{vertical\ advection} + \underbrace{\frac{\partial K_z \partial O_2}{\partial z^2}}_{vertical\ mixing}$$

where $u_H$ and $w$ are respectively the horizontal and vertical currents and $K_z$ is the vertical mixing coefficient computed by the dynamical model.

10 Finally, $J_{flux}$ is the air-sea flux of $O_2$ ($F_{O_2}$) divided by the depth of model surface layer, where

$$F_{O_2} = k_w(\alpha O_{2atm} - O_2)$$

$k_w$ is the transfer velocity, $(\alpha O_{2atm} - O_2)$ is the difference in $O_2$ partial pressure between the air and surface sea water, and α the solubility of $O_2$ in seawater.

The NEMO-PISCES coupled biophysical model has been successfully applied to various studies in the Indian Ocean (e.g. 15 Kone et al., 2009; Resplandy et al. 2009; Currie et al., 2013; Keerthi et al. 2016), including the Arabian Sea OMZ (Resplandy et al., 2011, 2012). A more detailed model description is provided in the manuals for NEMO and PISCES available online at http://www.nemo-ocean.eu/About-NEMO/Reference-manuals.

Here, we specifically use a ¼° resolution (i.e. cell size ~25 km) Indian Ocean regional configuration, which is a sub-domain of the global configuration described by Barnier et al. (2006). It has 46 vertical levels, with a resolution ranging from 5m at 20 the surface to 250m at the bottom. The African continent closes the western boundary of the domain. The oceanic portions of the eastern, northern and southern boundaries use radiative open boundaries (Treguier et al., 2001), constrained with a 150-day timescale relaxation to physical and biogeochemical inputs from a global simulation (Dussin et al., 2009). The circulation and thermodynamics of this regional configuration have been extensively evaluated and found to perform well in several Indian Ocean regions (Praveen Kumar et al., 2014), including the Arabian Sea (Nisha et al., 2013; Vialard et al., 25 2013; Keerthi et al., 2015, 2016) and the Bay of Bengal (Akhil et al., 2014, 2016).

The simulation starts from the rest and the WOA13 climatology temperature and salinity (Boyer et al., 2013). PISCES biogeochemical tracers are initialized from the WOA13 database for nutrients and a global simulation climatology for other

tracers (Aumont and Bopp, 2006). After 5 years of spin-up with climatological surface and lateral boundary forcing, the model is forced with the Drakkar Forcing Set #4.4 (DFS4.4, Brodeau et al., 2009) from 1958 to 2012. This forcing is a modified version of the CORE dataset (Large and Yeager, 2004), with atmospheric parameters derived from ERA40 reanalysis until 2002 (Uppala et al., 2005) and ECMWF analysis after 2002 for latent and sensible heat fluxes computation.

Radiative fluxes are taken from the corrected International Satellite Cloud Climatology Project-Flux Dataset (ISCCP-FD) surface radiations (Zhang et al., 2004), while precipitation are specified from a blend of satellite products described in Large and Yeager (2004). All atmospheric fields are corrected to avoid temporal discontinuities and to remove known biases (see Brodeau et al., 2009 for details). In the following, the 1960-2012 period is analyzed.

**2.3 Model climatology**

The model's ability to capture the climatological surface chlorophyll concentrations during the summer and winter monsoons is illustrated on Fig. 2. During summer monsoon (Fig. 2a), seasonal blooms are observed along the coasts of the Arabian Peninsula and along the WCI in response to coastal upwelling that brings nutrients into the euphotic layer (e.g. Wiggert et al., 2005; Levy et al., 2007; Koné et al., 2009). The chlorophyll signal along the Somalia and Omani coasts extends offshore towards the central Arabian Sea (Fig. 2a) through offshore lateral advection of nutrients from upwelling

regions, either by large-scale circulation or by eddy activity (e.g. Lee et al., 2000; Resplandy et al., 2011). During winter monsoon (Fig. 2b), the cool, dry northeasterly winds in the Arabian Sea induce convective mixing and entrain nutrient-rich waters to the surface, triggering the chlorophyll bloom observed north of 15°N (Madhupratap et al., 1996). The model generally accurately reproduces these seasonal chlorophyll patterns (Figs. 2c and 2d). As for observations, the largest chlorophyll bloom in summer occurs in the western Arabian Sea, with strong signals along the rim of the northern Indian

Ocean, while winter is characterized by oligotrophic conditions in the southeastern Arabian Sea and higher concentrations in the northern and western parts. The largest discrepancy between modeled and satellite chlorophyll is an overall overestimation of the amplitude of the summer blooms, in the western Arabian Sea, east of Sri Lanka and around the rim the Bay of Bengal (Figs. 2a and 2c). The seasonal chlorophyll patterns in fall are very similar, although weaker than those in summer in both observations and model (not shown).

The comparison of the modeled horizontal and vertical climatological oxygen distribution with that of WOA13 is shown in Fig. 3. In observations, the core of the OMZ is confined to the northern part of the basin (Fig. 3a) and expands vertically between 150 and 1000-m depth (Fig. 3b), with lowest subsurface oxygen concentrations found in the central/eastern part of the basin. The OMZ is thus shifted to the east of the region of highest biological production located along the west coast of the Arabian Sea (Fig. 2). The oxycline lies around ~100m, and is slightly shallower on the western and eastern part of the

basin (Fig. 3b), because of the seasonal upwelling systems there. The model captures these observed oxygen patterns, with poorly-oxygenated water confined to the northern Arabian Sea and high-oxygenated water found near the equator and farther south, and an OMZ core located in the eastern part of the Arabian Sea despite a slight overestimation of the modeled oxygen

content at depth in the Arabian Sea (Figs. 3b vs 3d). We can also note that the model underestimates the depth of the OMZ core (~200m in the model and ~300m in observations). The upper ocean vertical oxygen distribution is well captured, with a model oxycline depth around 100m (Fig. 3d), similar to the observed one (Fig. 3c).

## 2.4 Thermocline and oxycline depths

Both temperature and dissolved oxygen decrease with increasing depth below the mixed layer (Figs. 3b and d). The oxycline or thermocline depths are defined as the depths of maximum gradient. It is however customary to approximate those depths from a fixed isocontour, especially in tropical regions. Resplandy et al. (2012) used the depth of 100 $\mu$mol.L$^{-1}$ as a proxy for the oxycline depth in their study, whereas Prakash et al. (2013) used 50 $\mu$mol.L$^{-1}$. Here, we define the oxycline depth (hereafter OCD) as the depth of the 100 $\mu$mol.L$^{-1}$ oxygen isocontour (following Resplandy et al., 2012) and the thermocline

depth (hereafter TCD) as the depth of 23$^{o}$C isotherm (following Prakash et al., 2013). The conclusions of our study are not sensitive to the above choices in the northern Indian Ocean (and especially in our focus region along the WCI), i.e., the results discussed below are similar when considering the 50 $\mu$mol.L$^{-1}$ or 150 $\mu$mol.L$^{-1}$ instead of 100 $\mu$mol.L$^{-1}$, to define the oxycline depth and 20°C or 25°C instead of 23°C to define the thermocline depth. The OCD and TCD were both derived from observed or modeled profiles using linear interpolation. To ascertain the variability along the WCI, we average relevant

parameters over a box located between 10°N and 15°N extending from the coast to 2° offshore (hereafter referred to as the WCI box, shown on Fig. 5). It must be noticed that the results discussed below are not sensitive to any slight modifications of the box boundaries or offshore extent of this box. Interannual anomalies of all variables are calculated from monthly time-series by subtracting the mean seasonal cycle and applying a 3-month smoothing to remove the sub-seasonal variations.

## 2.5 IOD index

To characterize the IOD variability, we used the standard definition of the Dipole Mode Index (DMI, Saji et al., 1999), which is calculated as the difference between the interannual SST anomalies in the western (50°E-70°E; 10°N-10°S) and southeastern (90°E-110°E; 10°S-0°) equatorial Indian Ocean averaged over September–November. We have used the DMI derived from observed SST, but the very high correlation between the DMI based on model and observations (~0.94) make our results fairly insensitive to the choice of either model or observationally-based DMI. This index has been normalized by

its standard deviation to make it dimensionless.

## 3 Results

### 3.1 Physical control of the seasonal oxygen variability off the WCI

Parvathi V 2/3/2017 11:02 AM
**Deleted:** Dynamical control of the oxycline variability along the WCI

Matthieu Lengaigne 6/3/2017 1:39 PM
**Deleted:** S

Matthieu Lengaigne 6/3/2017 1:39 PM
**Deleted:** of oxygen

The seasonality of the oxygen concentration within the core of the OMZ is very weak, but this is not the case for the OCD (Resplandy et al., 2012). Fig. 4 shows the depth-time section of the oxygen and temperature seasonal cycle off the WCI (see black frame on Fig. 5 for the WCI box location) from both model and observations. This figure illustrates that the seasonal evolution of temperature and oxygen are very similar, with both OCD and TCD start shoaling in April (~ 100 m) and reaching their shallowest depth during September-October (~50 m). This figure thus indicates a tight coupling between the oxycline and thermocline depth variations off the WCI. Resplandy et al. (2012) performed a budget analysis of the terms contributing to the seasonal oxygen variability along the WCI and showed that, although the biological sink and the dynamical sources of oxygen compensate on annual average, the oxygen seasonality in this region primarily arises from the vertical advection of oxygen. Vertical velocities act in the same way on the oxygen and temperature isolines, lifting or lowering them in phase as illustrated in Fig. 4, resulting in strong correlation between OCD and TCD along the WCI.

The above close link between the thermocline and oxycline has also been observed in many regions (e.g., Morales et al., 1999; Prakash et al., 2013). Fig. 5 quantifies this relationship for the entire northern Indian Ocean by displaying the correlation between the seasonal OCD and TCD variations for both observations and model, and these correlations exceed 0.7 everywhere in the Arabian Sea, except in a small region off the Horn of Africa. These correlations drop in the equatorial region, presumably because our oxycline definition (100 $\mu$mol.L$^{-1}$) corresponds to a lower temperature criterion than 23°C (for defining TCD) in that region. The model in general exhibits higher OCD-TCD correlations than in the observations, especially in the eastern Bay of Bengal where observed OCD and TCD are not well correlated. The tight OCD-TCD relationship along the WCI implies a strong control of seasonal oxygen variability by the upwelling intensity in that region.

The strong dynamical control on the oxygen variability is further illustrated in Fig. 6, which displays spatial maps of observed and modeled OCD and TCD seasonal climatologies. During the spring inter monsoon (March-May), the TCD and OCD are spatially quite uniform and deep (~100 m) in the southeastern Arabian Sea (Figs. 6a and 6e). This is also the case along the WCI south of 15°N despite the local upwelling favorable alongshore winds, indicating a remote control of OCD and TCD variations there. In contrast, the shallower OCD/TCD near the Southern Tip of India (STI) during this season is consistent with upwelling favorable winds in this region (Figs. 6a and 6e). With the advent of the summer monsoon (June-August), the westerly monsoon winds drive a very strong offshore Ekman transport in the western Bay of Bengal and near the STI (Suresh et al., 2016), resulting in an upwelling signal, which shoals the OCD/TCD up to 60 m at the STI (Figs. 6b and 6f; Smitha et al., 2008; Gupta et al., 2016). Further north, the winds are almost perpendicular to the coast of western India (Suresh et al., 2016) and hence do not induce upwelling there. The shallow OCD/TCD signal from the STI propagates northward along the western Indian coast as an upwelling coastal Kelvin wave (Suresh et al., 2016; see Figs. 6b and 6f), with OCD/TCD patterns clearly suggestive of faster wave propagation at lower latitudes. By the end of the southwest monsoon (September-October-November, hereafter SON), shallow OCD/TCD can be seen along the entire Arabian Sea coastal rim, primarily resulting from the remotely-forced coastal Kelvin waves. From December to February, winds at the STI and

alongshore winds further north are both favorable to downwelling, thus leading to deepening of the TCD and OCD along the WCI (Figs. 6d and 6h). Part of the strong upwelling signal at the coast during the preceding season (Figs. 6c and 6g) is radiated westward as a planetary wave (Figs. 6d and 6h; e.g. McCreary et al., 1993, Suresh et al., 2016). The striking similarity in the seasonal patterns of OCD and TCD again indicates a strong physical control of the eastern Arabian Sea

OCD at the seasonal timescale. This echoes results of Resplandy et al. (2012), who found that upper-ocean oxygen seasonality primarily arises from vertical movements at the thermocline level forced by both local and remote wind forcing.

In order to further illustrate the remote forcing effects discussed above, and the connection between the offshore oxygen variability and that on the shelf, Fig. 7 shows the seasonal evolution of remote and local winds (Fig. 7a) and a comparison of the climatological near-surface oxygen contents (0-40m average) obtained from the model and WOA13 offshore to the WCI

(i.e. in the WCI box) with those obtained from the CaTS on the shelf (Figs. 7b and c). The modeled and WOA oxygen climatologies in the WCI box match very well for the entire year.  In contrast, the CaTs in-situ measurements show lower oxygen concentrations.  The oxygen concentration on the shelf is expected to be lower than that offshore because of a higher oxygen consumption on the shelf, particularly associated with sediment respiration process, which in our model is a very simple representation.  Despite lower oxygen content on the shelf, there is a very good phase agreement between the oxygen

seasonal variations from CaTS in-situ measurements and the corresponding offshore variations derived from the WOA and model, suggesting a strong offshore-shelf connection through dynamical upwelling process, as suggested by Banse (1959), Carruthers et al. (1959) and Gupta et al. (2016).

The seasonal wind evolution shown in Fig. 7a allows discussing the respective influences of local (WCI box) and remote forcing (at STI) in driving the seasonal oxycline variations along the WCI. As discussed above (e.g. Fig. 4), the upwelling

off WCI starts developing (reduction in upper ocean oxygen content) at the beginning of the summer monsoon (April-May), reaches a maximum (minimum upper ocean oxygen content) by September-October and decays (increase upper ocean oxygen content) by November. The local winds along the WCI (black continuous curve on Fig. 7a) are favorable to upwelling only during the southwest monsoon (i.e., they have an alongshore southward component only from June to August), which indicates that the upwelling along the WCI is to a large extent driven by remote winds (Shetye et al., 1990).

The wind near the STI is upwelling-favorable from about April to October, and hence matches the seasonality of the upwelling of cold and low-oxygen waters, in agreement with the results of Suresh et al. (2016).

### 3.2 Physical control of the interannual oxygen variability off the WCI

The gaps in the CaTS observational dataset do not allow constructing a reliable time series of interannual upper ocean oxygen content, to which our model could be validated. However, the data can still provide some estimate of the amplitude

of interannual variability, which can be compared to that of our model (the whiskers on Figs. 7b and 7c indicate the amplitude of the variability around the mean seasonal cycle). Despite a slight underestimation in our model, the amplitude of

the near-surface oxygen interannual variability is largest during SON, both in the shelf observations and the model. This further corroborates the offshore-shelf connection discussed above.

A seasonally shallow OCD combined with a larger interannual variability in fall creates a window of opportunity for the occurrence of coastal anoxic events. Figs. 7d and 7e display the monthly percentages of occurrence of hypoxic profiles from
CaTS (on the shelf) data and the model (offshore). While the general patterns of oxygen/oxycline variability on the shelf and offshore to the coast remain similar, the actual upper ocean oxygen content and vertical oxygen profiles are different. Hence, we have used different thresholds to detect hypoxic profiles in the observation and the model. Consistent with previous literature, anoxic events are most likely to occur from August to November in the model and the shelf data, as expected from the very shallow background oxycline at that time of the year. This justifies our focus on the fall period for analyzing the
processes that drive the modeled interannual variability of the WCI oxycline in the following.

We previously demonstrated a tight relationship between the seasonal variability of OCD and TCD in the eastern Arabian Sea. Fig. 8a exhibits a similar relation in large portions of the northern Indian Ocean for fall interannual OCD and TCD anomalies in the model. A comparison with observations is unfortunately not feasible due to lack of a basin-scale dataset for interannual OCD anomalies. The correlation between interannual OCD and TCD anomalies are, in general, slightly weaker
than that at seasonal timescales (Fig. 5b), but remain high in a large part of the Indian Ocean north of 5°N, generally exceeding 0.8 in the entire Bay of Bengal and in the eastern Arabian Sea, and in particular in the WCI box (~0.95 correlation, Fig. 8b).

The influence of remote forcing at WCI is further established in Fig. 9. Due to unavailability of the continuous oxygen observations (see Section 2.1), we cannot directly evaluate the modeled oxygen interannual variability in the WCI region.
We however evaluate the modeled TCD interannual variability, which is closely tied to the OCD interannual variability (~0.95 correlation, Fig. 8b). The modeled interannual TCD anomalies in the WCI box (red curve on Fig. 9a) agree well (0.84 correlation) with sea-level interannual anomalies (a good proxy for TCD variations in stratified regions) from altimeter measurements (blue curve on Fig. 9a) during fall. Despite instances, when the model agreement is weaker, like during 2002-2006 period, both the modeled TCD and the observed sea level indicate strongest thermocline shoaling in fall 1999 and
deeper than usual thermocline in fall 1994, 1997 and 2008. We will exploit this ability of our model to capture the observed TCD interannual variations along the WCI to further examine the processes responsible for the OCD interannual variability. Along with fall TCD and sea-level anomalies at WCI, Fig. 9a also displays the fall interannual anomalies of zonal winds at the STI (black dashed curve on Fig. 9a; box with dashed frame in Figs. 9b and 9c). The good phase agreement between the interannual fluctuations of zonal winds at STI and both modeled TCD and observed SLA (-0.65 and -0.69 correlation
respectively, Fig. 9a) at WCI is a strong indication that the interannual variations along the WCI are strongly influenced by the wind variations at the STI, through coastal Kelvin waves propagation, as demonstrated by Suresh et al. (2016) for the

seasonal timescale. Fig. 9b shows the correlation of interannual anomalies of model TCD everywhere in the northern Indian Ocean with the time series of model TCD in the WCI box during fall. Fig. 9c shows a similar analysis, but for the observed SLA. In both model and observations, the thermocline depth variations along the WCI in fall are associated with basin-scale coherent signals in thermocline depth, sea level and wind stresses. Those basin-scale patterns are very similar to those

associated with a positive phase of the IOD mode, with upwelling in the Eastern Equatorial Indian Ocean (EEIO) and a downwelling in the western Indian Ocean. In the following section, we will show how IOD events influence the fall thermocline and oxycline depths off the WCI.

The relationships between the modeled interannual variability of the OCD along the WCI and that of the OCD, TCD, SST and wind at the basin scale are demonstrated in Figs. 10a and 10b, which show regression maps of fall interannual anomalies

of these variables to the time series of the fall WCI OCD anomalies shown in Fig. 8b (black line) over the 1960-2012 period. These maps display the typical basin-scale anomalies corresponding to an anomalously deep OCD in the WCI. Consistent with Fig. 9, the fall WCI OCD variations are not merely local, but are associated with basin-scale ocean-atmosphere interannual anomalies over the entire equatorial and northern Indian Ocean. An anomalously deep OCD off the WCI is usually associated with deeper OCD and TCD (i.e. positive anomalies) in the southeastern Arabian Sea and in the vicinity of

Sri Lanka and the STI (Figs. 10a and 10b). Positive OCD anomalies off the WCI are also related to shallower OCD and TCD in the eastern Indian Ocean and along the eastern rim of the Bay of Bengal (i.e. negative anomalies). The associated large-scale wind patterns (Fig. 10a) explain these interannual OCD and TCD patterns. Anomalous easterlies in the equatorial band force upwelling equatorial Kelvin waves that shoal the OCD and TCD in the EEIO. These signals further propagate around the rim of the Bay as upwelling coastal Kelvin waves, thereby shoaling the TCD and OCD there (e.g. McCreary et al., 1993;

McCreary et al. 1996; Aparna et al., 2012). Similar to what happens at the seasonal scale (Suresh et al., 2016), easterly zonal wind stress anomalies in the vicinity of Sri Lanka and the STI force a downwelling coastal Kelvin wave that propagates poleward along the western Indian coastline, resulting in deepening of the TCD and OCD there. The strong negative correlation between the WCI OCD and STI zonal winds interannual fluctuations (-0.73 over the entire period; see Fig. 11b) further illustrates the strong influence of these winds in driving OCD and TCD interannual fluctuations in the southeastern

part of the Arabian Sea. In contrast, the local alongshore wind variations along the WCI have a weaker influence on the WCI OCD, with a negative correlation of -0.25 (Fig. 11c), confirming the dominance of remote forcing from the STI region over the influence of local winds.

### 3.3 The specific role of the IOD

The SST variations associated with the OCD signals in the WCI region are characterized by a clear signal in the EEIO (near

the Sumatra coast), and weaker signals of opposite sign in the western Indian Ocean. As pointed out before, the patterns shown on Figs. 9b, 9c and Fig. 10a, 10b are reminiscent of the IOD signature (Saji et al., 1999; Webster et al. 1999;

Murtugudde et al. 2000), an Indian-Ocean coupled ocean-atmosphere climate mode that peaks in fall, as discussed in the introduction. This is further demonstrated in Figs. 10c and 10d, which display regression maps of interannual anomalies of OCD, TCD, SST, and winds onto the boreal fall DMI. The resulting patterns, representing the typical anomalies associated with a positive IOD phase, are strikingly similar to those displayed in Figs. 10a and 10b (pattern correlation of ~0.85). This highlights the strong link between the IOD events and the WCI fall oxycline year-to-year variations. We further examine the relationship between the WCI OCD and the IOD in Fig. 11a, which displays time series of fall OCD interannual anomalies and the fall DMI (computed from both modeled and observed SST, the correlation between them being 0.94). Consistent with the regression map, we find a high correlation between the WCI OCD and the DMI (~0.67/0.62 with modeled and observed DMI respectively).

The influences of positive and negative phases of IOD on the WCI OCD (and TCD, not shown) are however not symmetrical (Fig. 11a): most of the positive IOD events cause a deepening of OCD along WCI (e.g., 1961, 1967, 1994, 1997), while negative IOD can either be associated with a shoaling (e.g. 1996, 1998, 2010) or even a deepening (e.g. 1979-81). To further illustrate this asymmetry, Fig. 12 provides a scatterplot of the fall DMI versus the fall interannual anomalies of OCD off the WCI. This scatterplot confirms that there is an asymmetric impact of positive and negative IODs on the OCD along the WCI. The WCI OCD response to positive IODs is very robust, with all positive IOD events (DMI > 1) except one being systematically associated with a deepening. The WCI response to negative IOD is weaker and much less systematic. As discussed above, negative IOD events (DMI<-1) are generally related to negative OCD anomalies but can also be related to positive OCD anomalies (Fig. 11a). Fig. 13 provides a composite of the temporal evolution of the anomalous OCD and TCD along the WCI along with zonal wind stress variations at the STI for signal associated with positive/negative IOD events. It must be noticed here that time series for negative IOD events has been multiplied by -1 to ease comparison with positive events. Figs. 13a and 13b illustrate again the two to three times weaker response of the WCI OCD for negative compared to positive IOD events. This weaker response is related to slightly weaker zonal wind anomalies at the STI (Fig. 13c) that consequently trigger a weaker coastal Kelvin wave response along WCI and thus weaker WCI TCD/OCD anomalies. It is therefore likely that the weaker and less robust wind signal associated with negative IODs at the STI compared to positive IODs may explain part of the asymmetry seen in Fig. 12. We will discuss the possible causes for this asymmetry in section 4.2.

**4 Summary and Discussion**

**4.1 Summary**

The year-to-year variations of coastal hypoxia along the west coast of India (WCI) have been identified by Naqvi et al. (2000, 2009), along with their strong impacts on fisheries and ecosystem. The mechanisms controlling these variations have however not yet been elucidated. The present study offers new insights on the physical controls of coastal hypoxia along the

WCI. To that end, we used an eddy-permitting (¼° horizontal resolution), regional Indian Ocean configuration of a coupled physical-biogeochemical model. The simulation spans a period long enough (1960-2012) to allow analyzing the driving mechanisms of interannual variability of oxygen off the WCI. The model accurately reproduces the oxycline and thermocline seasonal cycle off the WCI, with a seasonal upwelling that yields shallowest oxycline/thermocline at the end of the summer monsoon. The modeled and observed offshore climatological seasonal cycles of oxygen match in-situ measurements on the shelf, with a strongest seasonal oxygen deficiency and highest occurrence rate of anoxic events during boreal fall. It is suggested that the upwelling of oxygen-depleted subsurface waters at the shelf break influences the occurrence of anoxic events over the western Indian continental shelf.

The shallow oxycline in fall combines with a large interannual variability at this time of the year to create a window of opportunity for coastal anoxic events. Our model analysis further indicates that there is a tight coupling between the thermocline and oxycline variability in this region at both seasonal and interannual timescales, indicative of a strong physical control of the oxygen variability through vertical advection. Interannual thermocline fluctuations along the WCI are related to basin-scale wind, thermocline and oxycline depth perturbations associated with IOD events, an Indian Ocean coupled ocean-atmosphere climate mode that peaks in fall. Positive IOD events are associated with easterly wind anomalies in the central equatorial Indian Ocean, that extend meridionally up to the southern tip of India. These easterly wind anomalies trigger downwelling coastal Kelvin waves that propagate along the WCI and deepen the thermocline and oxycline in boreal fall, thereby preventing the occurrence of coastal anoxia off the WCI during positive IOD events. Our model results also suggest an asymmetry between the impact of positive and negative IOD events on the WCI oxycline depth. The westerly wind anomalies at the southern tip of India indeed have a smaller amplitude during negative IODs than their easterly counterparts during positive IODs, thus resulting in a weaker and less consistent shoaling of oxycline/thermocline along WCI during negative IOD events.

## 4.2 Discussion

### 4.2.1 Influence of the IOD on the interannual oxygen variability along the WCI

Previous studies have demonstrated the impact of large-scale climate modes on year-to-year variations of the oxygen deficiencies in coastal hypoxic systems elsewhere in the world ocean. In the Pacific, El Niño conditions lead to intensified oxygenation along the coasts of Peru and Chile as a result of weak upwelling (e.g., Arntz et al., 2006; Gutierrez et al., 2008), while in the Atlantic, the Benguela Niño leads to intensified anoxia along the Namibian shelf (Monteiro et al., 2008). The western continental shelf of India is home to the largest naturally-formed coastal hypoxic system in the world. In this study, we identify, for the first time, the IOD as the major climatic driver of the year-to-year oxycline and thermocline variations offshore to the WCI. Though the IOD has a weaker thermocline depth signature on the west than on the east coast of India, it has stronger societal consequences as it influences the WCI seasonal upwelling that brings suboxic waters very close to the

surface during fall. Although the IOD influence on the west Indian coast has never been reported so far, it has regularly been reported in the in the Bay of Bengal. In line with our results, Aparna et al. (2012) indeed showed that IOD events drive strong sea-level and thermocline fluctuations along the rim of the Bay in fall, through coastal Kelvin wave propagation from the equatorial region. Akhil et al. (2016) further demonstrated that this remote forcing also drives anti-clockwise anomalous

horizontal currents in fall in the Bay, which in turn leads to large interannual variations of Sea Surface Salinity in the southern Andaman Sea. On the biogeochemical side, Wiggert et al. (2002, 2009) and Currie et al. (2013) demonstrated that IOD events are responsible for large interannual chlorophyll variations in the southeastern Bay of Bengal and at the STI. Finally, the IOD signature found in Arabian Sea in the present study has already been described for the Bay of Bengal in terms of sea-level (Aparna et al., 2012) and chlorophyll (Currie et al., 2013).

The influence of IOD is further shown to be larger for positive than its negative phase. Our results suggest that part of the weaker WCI oxycline depth response during negative IOD may be explained by the weaker windstress anomalies at the STI associated with negative IOD events. This weaker wind amplitude could simply be related to the tendency of negative IOD events to be weaker than their positive counterpart (Saji and Yamagata 2003; Hong et al. 2008; Cai et al. 2013) or to asymmetries in the spatial patterns of winds associated associated with the non-linear response of deep atmospheric

convection to SST anomalies of each phase of the IOD. A more precise understanding of this asymmetry would require an in-depth investigation of the processes that control the wind variations at the STI and the thermocline along the WCI in response to positive and negative IOD events.

Our findings partly explain the substantial year-to-year changes in both the duration and intensity of the observed seasonal

oxygen deficiency over the western Indian shelf (Naqvi et al., 2009). None of the anoxic events reported by Naqvi et al. (2009) (black stars in Fig. 12) lies on the upper right quadrant of the scatterplot shown in Fig. 12, indicating that positive IODs systematically prevent the occurrence of anoxic events. For instance, the relaxation of anoxic condition in early fall 1997 reported by Naqvi et al. (2009) is in line with the occurrence of very strong positive IOD during that year. Most anoxic events are found in the lower left quadrant, i.e., near neutral or negative IOD conditions and anomalously shallow offshore

oxycline. Neutral or negative IOD years are however not necessarily anoxic, indicating that a neutral or negative IOD is a necessary but not a sufficient condition for severe anoxia. A recent study by Gupta et al. (2016) revealed that the oxygen deficiency in 1959 along the WCI was more severe than in 2012, a conclusion consistent with the occurrence of a negative IOD in 1959 and a positive one in 2012. Similarly, in-situ measurements also revealed that subsurface oxygen concentrations were significantly lower at the turn of the 20[th] century than in the 70's (Naqvi et al., 2009): our simulation exhibit a similar

behavior (see Fig. 11a), showing many years with shallower than normal OCD in the later period and systematically deeper than normal OCD during 1970's. The causes for those decadal variations need to be investigated in greater detail.

Matthieu Lengaigne 6/3/2017 1:24 PM
**Moved up [1]:** 4.2.1 Explanation to the observed year to year low oxygen condition variability along WCI

The ~0.7 correlation between IOD variability and oxycline variations along the WCI implies that ~50% of the interannual oxycline variance is explained by the IOD at this location. This relationship between the IOD and year-to-year variations of seasonal anoxic conditions along the shelf may facilitate advance warning for the possible occurrence of severe anoxic events. Recent studies indeed indicate that skillful predictions of mature IOD events in fall can be achieved about one season ahead (e.g. Luo et al., 2007; Wang et al., 2009; Zhao and Hendon 2009; Sooraj et al., 2012) and up to two seasons ahead in the case of large IOD events (Luo et al., 2007, 2008; Shi et al., 2012). Those predictions of IOD events should allow providing a warning about the likelihood of severe anoxic conditions along the WCI during spring or summer. A predicted positive IOD is indeed associated with very low chances of such an anoxic event, while neutral or negative IOD conditions may be associated with the occurrence of such an event.

It must however be kept in mind that other factors are also likely to contribute to the reported interannual fluctuations of hypoxic conditions in this region. Naqvi et al. (2009) for instance suggested that increased productivity due to increased nutrient loading from land associated with anthropogenic activities might have the potential to trigger a shift from natural suboxic to anthropogenic anoxic conditions during recent decades. This hypothesis however cannot explain the relaxation of the intensity of oxygen deficiency in the recent decades. Another contributing factor could be related to changes in local hydrographic variations. For instance, interannual variations of the land runoff along the Western Ghats, local precipitation during summer monsoon or input of Bay of Bengal freshwater during the northeast monsoon (e.g. Jensen et al., 2001) could modulate the upper ocean haline stratification, ventilation of the subsurface waters and hence the subsurface oxygen content along the WCI. Finally, local alongshore wind variations may modulate the intensity of coastal upwelling and hence the amount of oxygen-depleted waters brought to the shelf. The influence of these factors hence requires further investigation.

### 4.2.2 Model limitations

An obvious limitation of the current study is the spatial resolution of our model (~25 km). While our model has a reasonable representation of the temperature and oxygen seasonal variations in the deep ocean off the shelf break, its spatial resolution is not sufficient to resolve the details of physical processes controlling the upwelling along shelf break (e.g. Huthnance 1995; Allen et al. 2010). For the case of the narrow continental shelf along the west coast of North America, several studies have shown that a minimum horizontal resolution of 10 km is required (Marchesiello et al., 2003; Veneziani et al., 2009). For example, at ¼° resolution our model is eddy-permitting, but not eddy-resolving, and hence does not fully capture oceanic mesoscale eddies, which play an important role for the exchanges between the shelf and the open ocean in upwelling regions (e.g. Marchesiello et al. 2003 Bettencourt et al. 2015; Vergara et al. 2016). Another limitation arises from the absence of tidal forcing in the model, which may play an important role, as strong internal tides can be generated on the shelf break and contribute to enhance the thermocline vertical excursion and mixing, which can both contribute to bring more deep ocean oxygen-deficient water to the shelf (Monteiro et al. 2005).

While our model does not reproduce the details of exchanges between the shelf and open ocean, we have just used it as a proxy of the behaviour of open ocean, off the WCI. Several studies have already pointed towards the influence of offshore oxygen variations in driving the variability of hypoxic conditions along other coastal regions (e.g. Grantham et al. 2004; Helly and Levine, 2004; Arntz et al., 2006; Gutierrez et al., 2008). As was shown in Figs. 6 and 7b, the model and WOA climatology vertical oxygen distribution agree quite well, both in terms of the oxycline depth and near-surface value. The CaTS data, on the other hand, is representative of what happens much closer to the coast and displays much lower oxygen levels than seen further offshore in WOA and the model. This may of course partially be due to shortcomings in representing physical exchanges between the shelf and open ocean at the current resolution of our model and existing oxygen dataset in the region. But biological processes are also known to be a prominent oxygen consumption factor on the shelf, in particular in the benthic zone where the enhanced concentration of particulate matter above sediments is associated with high oxygen demand (e.g. Cowie, 2005). The crude parameterization of sediments in the model probably does not consume enough oxygen very close to the coast. On the other hand, the good phasing between the oxygen seasonal variability offshore (in the model and WOA) and shelf (CaTS) data (Figs. 7b and 7c) suggests that the offshore variability is probably an important driver of the oxygen content on the shelf. However, a proper representation of benthic biological processes would probably be needed to represent the low oxygen levels very close to the coast (Fig. 7c). Dedicated studies at higher spatial resolution with sensitivity tests on the representation of near-shore biological processes will probably be needed in order to better understand how the representation of near-shore biological processes constrain the coastal oxygen representation.

### 4.2.3 Observational requirements

On the observational front, the current spatio-temporal sampling does not allow building reliable long-term time series of the month-to-month oxygen variations along the shelf and offshore. Despite the establishment of frequent measurements of oxygen profile off Goa since September 1997, the numerous unsampled months (July and August are almost unsampled because of rough weather conditions) and the strong sub-monthly variability prevent a continuous monitoring of oxygen variations along the WCI. A reasonable number of moorings or Argo drifters with oxygen and temperature sensors along the shelf and further offshore would allow a finer description of the oxygen variability, its relationship with temperature and connection with the offshore variations. In order to establish an unequivocal evidence for the shelf-open ocean interactions, future studies should also consider improved observations such as repeated glider transects or triad of moorings (shelf, shelf break and open ocean) monitoring both physical and biogeochemical quantities in this region.

### 4.2.4 Hypoxia in other regions of the northern Indian Ocean

Though the present study is focused on the WCI, our Indian Ocean configuration model allows assessing other regions where near-surface hypoxia can occur. Fig.14 shows the percentage of profiles where oxygen concentrations below 80 µmol.l$^{-1}$ occur within the top 50 m. This threshold is indicative of the limit under which many organisms start to suffer from

physiological stress that could ultimately lead to death (Vaquer-Sunyer and Duarte, 2008). This analysis indicates that the coast of Oman can also experience hypoxic conditions as reported in the literature (e.g., Piontovski and Al-Oufi, 2015), although hypoxia along Oman is never as severe as that off WCI (Naqvi et al., 2010b). Fig. 14 indicates that the northwestern Bay of Bengal can also experience near-surface low-level oxygen concentrations, as reported from a series of
ship cruise measurements by Sarma et al. (2013). Further examination of the mechanisms driving these hypoxic events reveal that the IOD strongly impacts the oxygen variability in the northwestern Bay of Bengal: positive IOD events generally inducing a shoaling of the oxycline in this region (see Fig. 10c) through upwelling coastal Kelvin wave propagation from the equatorial region. In contrast, the influence of IOD along the Omani coast is almost negligible (Fig. 10c), and oxygen variations here seem to be related to offshore Ekman pumping (not shown). Further dedicated studies are needed to better
understand the oxygen variability in these sensitive regions and their potential impacts on the ecosystem and fisheries.

**Acknowledgements**

This work is a part of CSIR-funded INDIAS IDEA project. I. Suresh acknowledges financial support from Council of Scientific and Industrial Research, New Delhi and INCOIS/MoES (HOOFS program). V. Parvathi is funded by CSIR under Senior Research Fellowship. We thank CSIR-NIO data centre and Chemical Oceanography Division for making the archived
cruise and CaTS data available for the present study. We thank the NEMO-PISCES modeling team. The simulations were performed on HPC Pravah at CSIR-NIO. We thank M. Afroosa for assistance in data processing. M. Lengaigne, C. Ethé, J. Vialard and M. Levy benefited from Institut de Recherche pour le Développement (IRD) funding for their visits to the CSIR-NIO. This is NIO contribution number XXXX.

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

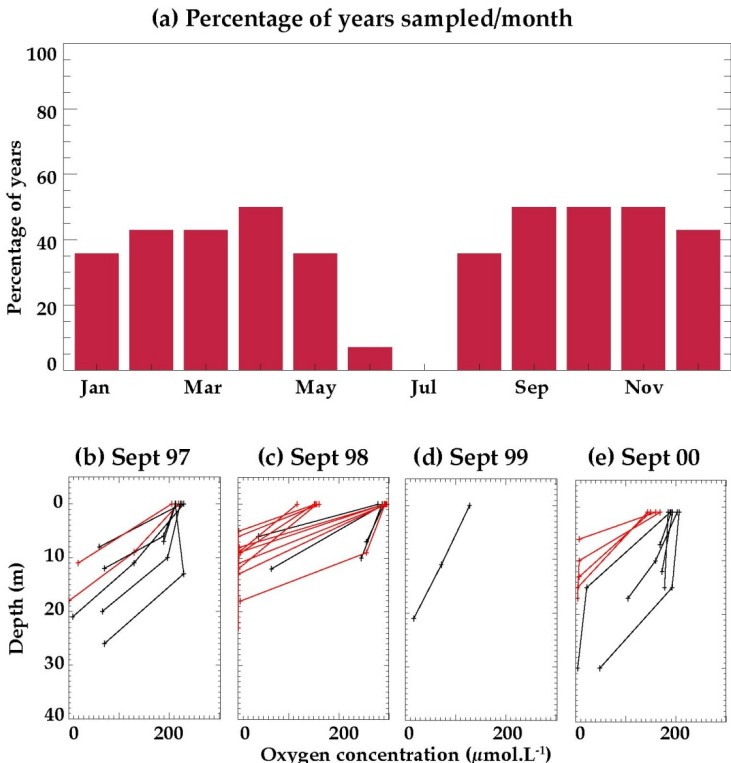

**Figure 1. (a)** Histogram of the percentage of years sampled for each calendar month over the 1997-2010 period at CaTS station. Vertical oxygen profiles collected at CaTS in September **(b)** 1997, **(c)** 1998, **(d)** 1999 and **(e)** 2000. Profiles with oxygen concentrations below 20 µmol.l[-1] within the top 20 m of the water column are indicated in red.

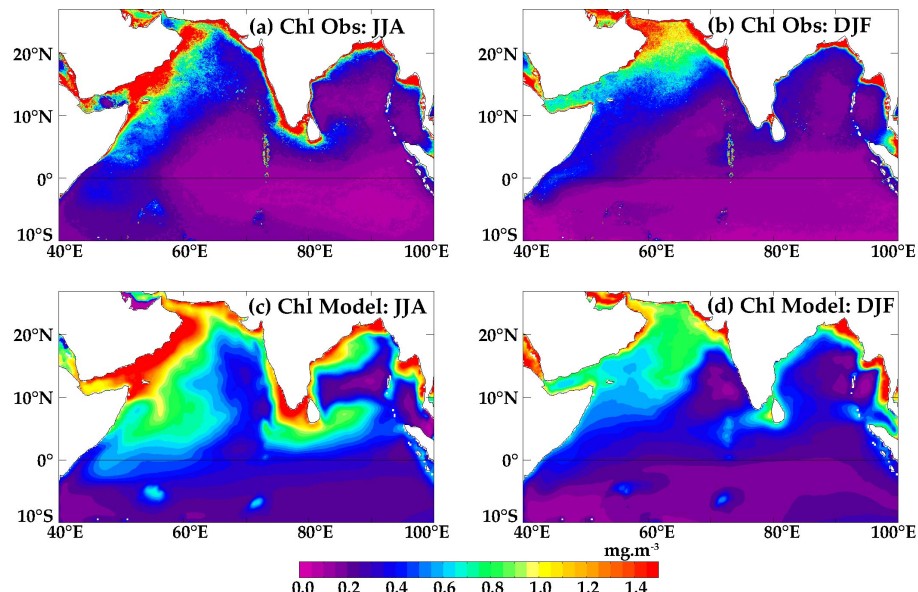

**Figure 2.** Northern Indian Ocean surface chlorophyll (mg.m$^{-3}$) climatology for (left) summer (June-August) and (right) winter (December-February) in (top) the ESA satellite product and (bottom) model.

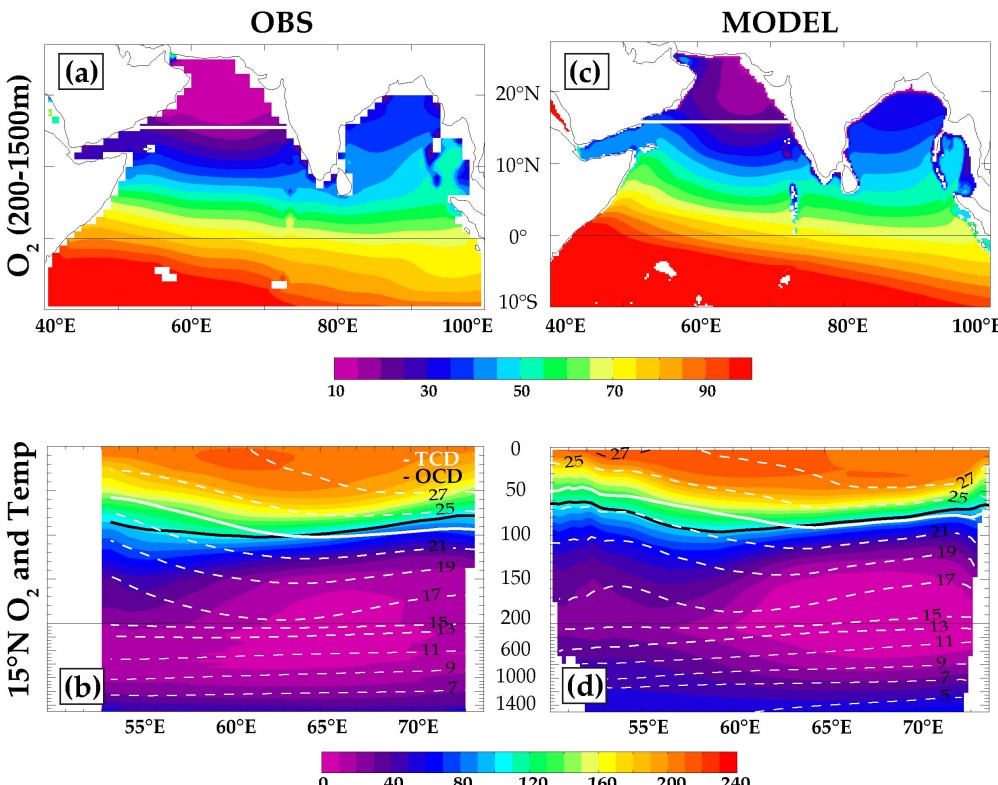

**Figure 3. (a)** Map of yearly climatological $O_2$ concentration ($\mu$mol.l$^{-1}$) averaged between 200 and 1500-m depths in the northern Indian Ocean from WOA13. **(b)** the vertical distribution of yearly climatological $O_2$ ($\mu$mol.l$^{-1}$; shaded) and temperature ($^\circ$C; dashed white contours with 2$^\circ$C interval) across East-West section at 15$^\circ$N (indicated by white line in panels a and c) in the Arabian Sea from WOA13 data. **(c, d)** Same as **(a, b)** but for the model. Depths of 23$^\circ$C isotherm (white line) and of 100 $\mu$mol.l$^{-1}$ isoline (black line) are marked on panels **b** and **d.**

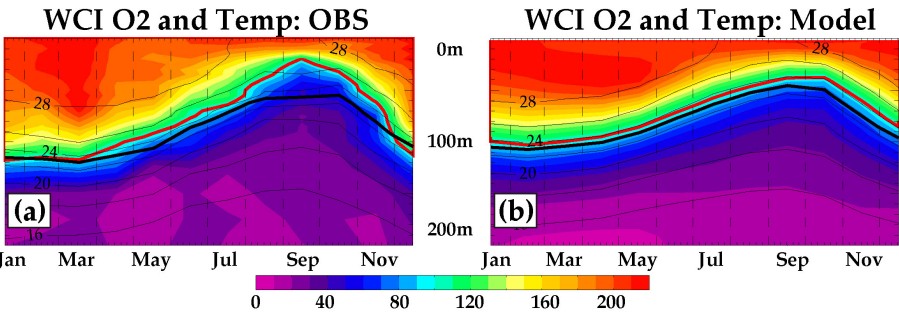

**Figure 4.** Seasonal evolution of oxygen (μmol.l⁻¹; color shaded) and temperature (°C; thin black contour) vertical profiles averaged over the WCI box (indicated as black frame on Fig. 5) from (a) WOA13 and (b) model . The oxycline and thermocline depths are marked by thick red and black lines respectively in both panels.

**Seasonal oxycline and thermocline depth correlation**

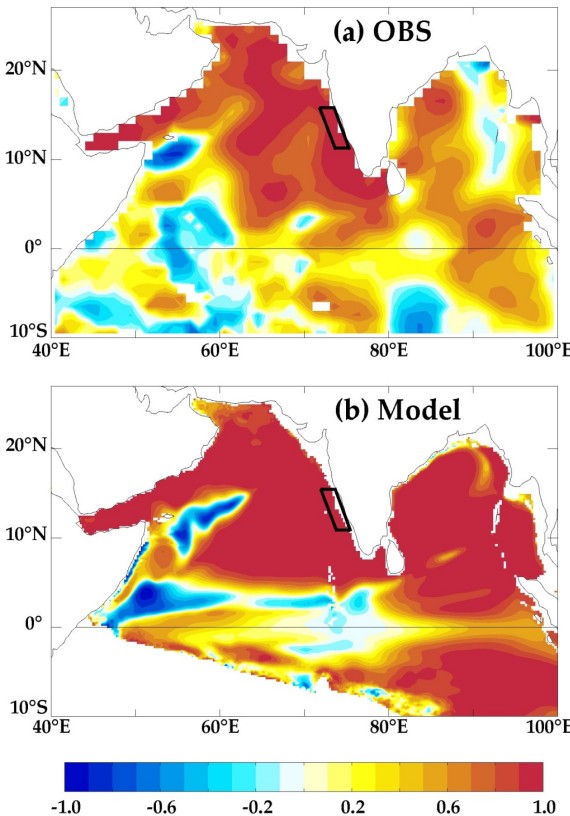

**Figure 5.** Maps of correlation between the mean seasonal cycle of oxycline and thermocline depths over the northern Indian Ocean from **(a)** WOA13 and **(b)** model. Values are masked when the oxycline could not be defined, i.e., when the oxygen concentration is above 100 μmol.l$^{-1}$ in the entire water column. The WCI box is marked as a black frame on each panel.

## Seasonal oxycline and thermocline depth

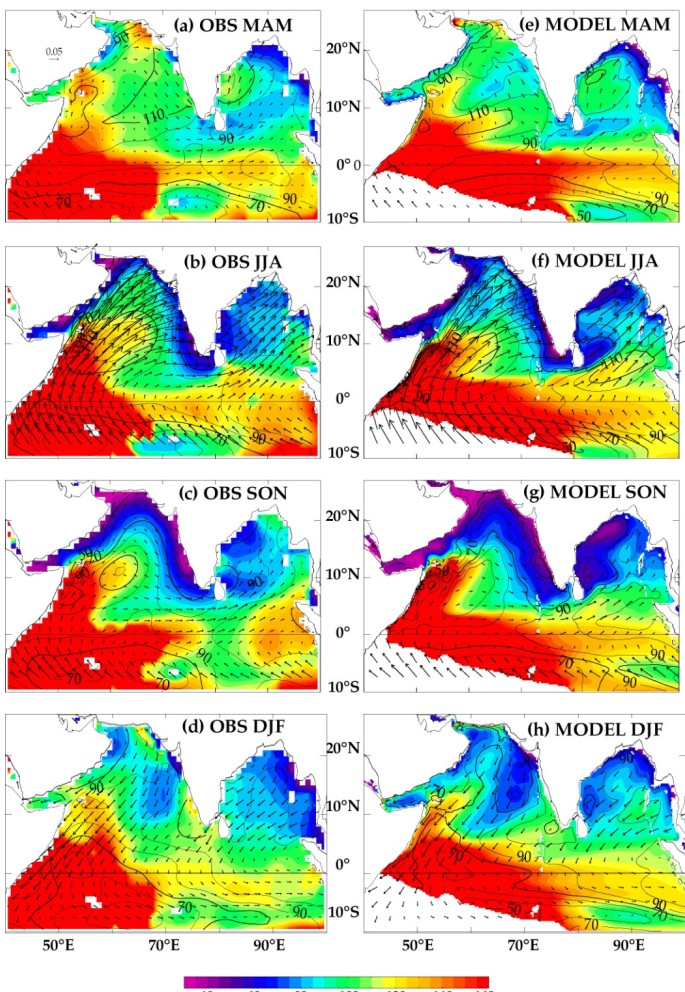

**Figure 6.** Maps of the seasonal climatology of oxycline (m; shaded) and thermocline depths (m; contours with 10-m interval) from WOA13 during **(a)** March-May, **(b)** June-August, **(c)** September-November and **(d)** December-February. **(e-h)** Same as **(a-d)**, but from the model. Seasonal wind stress $(N.m^{-2})$ patterns from the model forcing field are also shown as vectors on all panels.

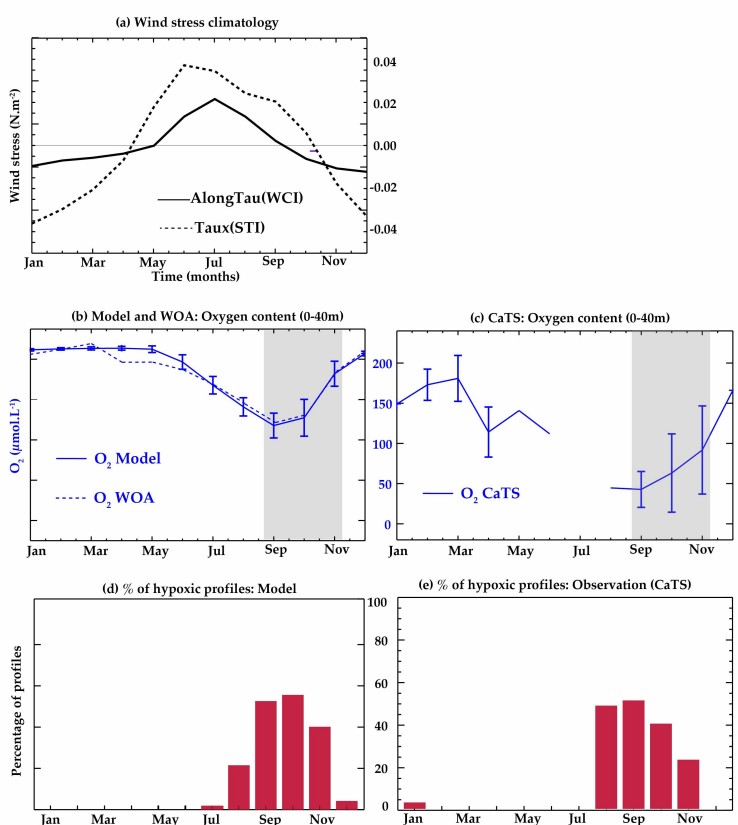

**Figure 7. (a)** Monthly climatological time series of alongshore wind stress in the WCI box (thick black curve) and zonal wind stress at the southern tip of India (STI; dashed black curve) with positive values denoting the upwelling-favourable winds. Monthly climatological time series of upper ocean (0-40 m depth) averaged oxygen (blue) **(b)** from the model (continuous) and WOA13 (dashed) in the WCI box and **(c)** from the in-situ CaTS data. Vertical bars on panel b and c indicate +/- one standard deviation around the mean value (displayed for CaTS data only when the number of years sampled for a given month exceeds five). Percentage of profiles for each calendar month for which **(d)** oxygen concentrations below 80 $\mu$mol.l$^{-1}$ occur within the top 50 m at WCI box in the model and **(e)** oxygen concentrations below 20 $\mu$mol.l$^{-1}$ occur within the top 20 m in CaTS data.

**(a) Interannual oxycline and thermocline depth correlation**

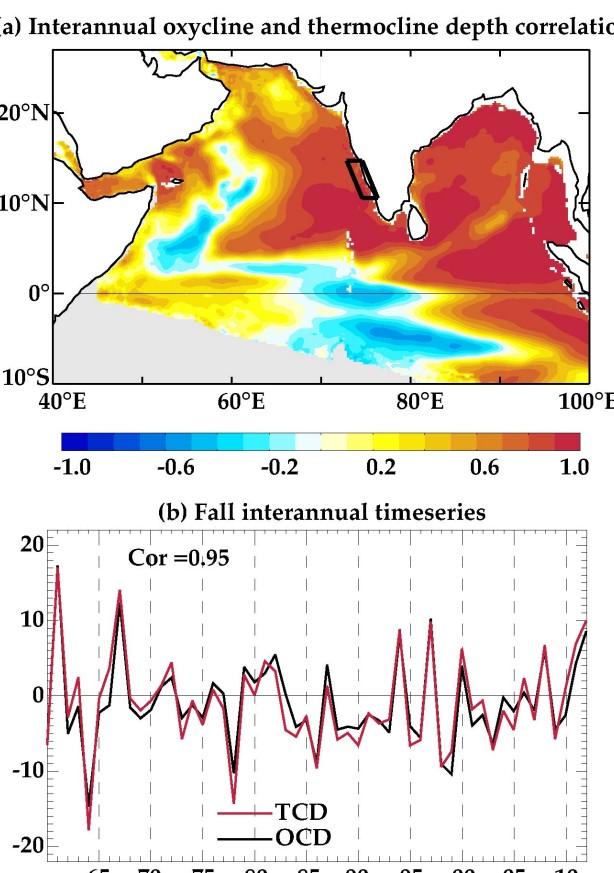

**(b) Fall interannual timeseries**

**Figure 8. (a)** Map of correlation between the modelled oxycline and thermocline depth fall interannual anomalies. **(b)** Time series of fall interannual anomalies of the modeled TCD (m; red line) and OCD (m; black line) averaged over the WCI box (see frame on panel a). On panel (a), values are grey shaded when the oxycline and/or thermocline could not be defined (with their present definition) for more than 20% of the profiles at a given location.

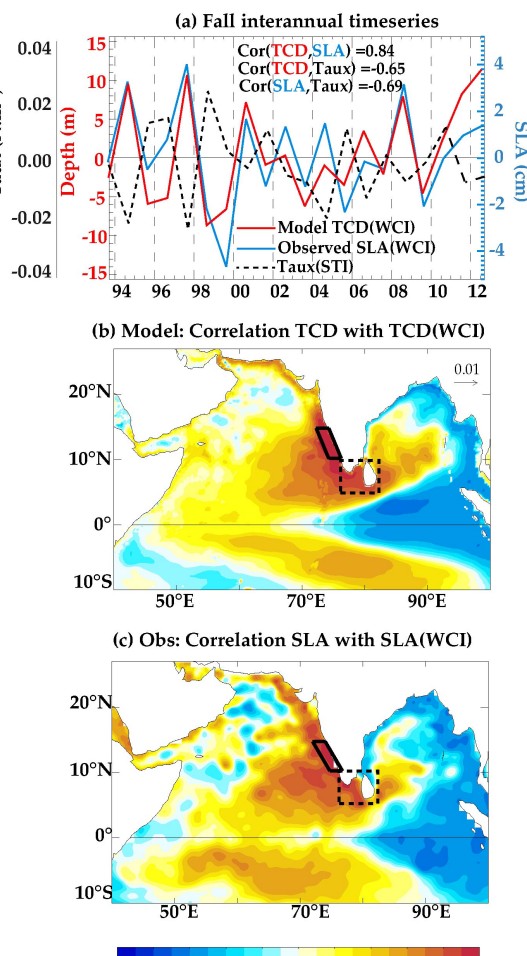

**Figure 9. (a)** Fall interannual anomalies of the model thermocline depth (red line) and the altimetry-derived sea-level (blue line) averaged over the WCI box (see black frame on panels b and c) along with fall interannual zonal wind stress anomalies (dashed black line) averaged over the STI box (see dashed frame on panels b and c). **(b)** Correlation pattern of fall interannual thermocline anomalies on to that averaged over the WCI box in the model. **(c)** Same as (b) but for altimetry-derived sea-level anomalies.

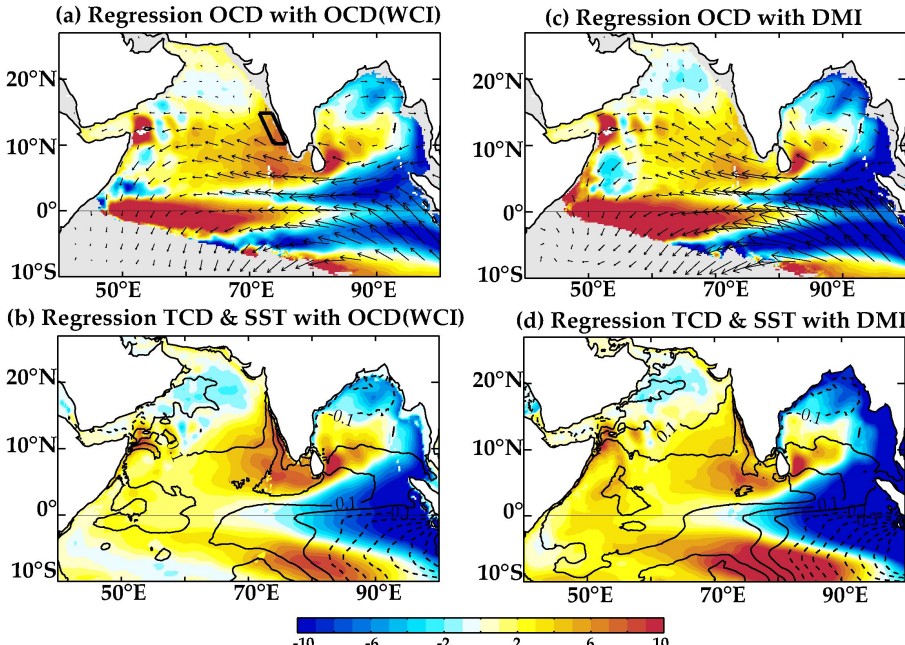

**Figure 10**. Regression patterns of fall interannual anomalies of modeled **(a)** oxycline (m; shaded) and wind stress (N.m$^{-2}$; vectors), **(b)** thermocline (m; shaded) and SST ($^{o}$C; contours with 0.1°C interval) onto the fall oxycline interannual anomalies averaged over WCI box normalized by its standard deviation. **(c-d)** Same as **(a-b)** but regressed onto the observed fall DMI index.

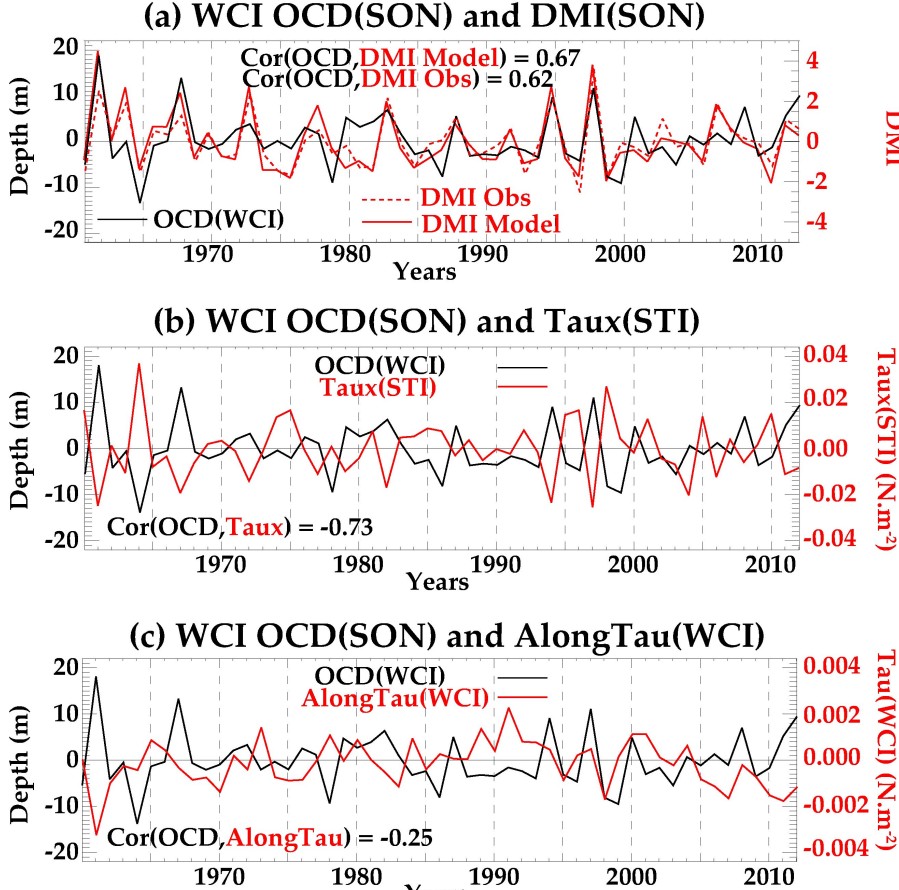

**Figure 11. (a)** Time series of fall interannual anomalies of modeled oxycline depth (continuous black line) averaged over the WCI box and modeled (continuous red) and observed (dashed red) fall DMI, **(b)** Time series of fall interannual anomalies of modelled oxycline depth averaged over the WCI box (black line) and zonal wind stress averaged over the southern tip of India (red line; STI box shown as dashed frame on Fig. 9b). **(c)** Time series of fall interannual anomalies of modelled oxycline depth (black line) and alongshore wind stress (red line) averaged over the WCI box. Upwelling-favourable winds on panel b and c are positive. Correlation coefficients between the variables are indicated in each panel.

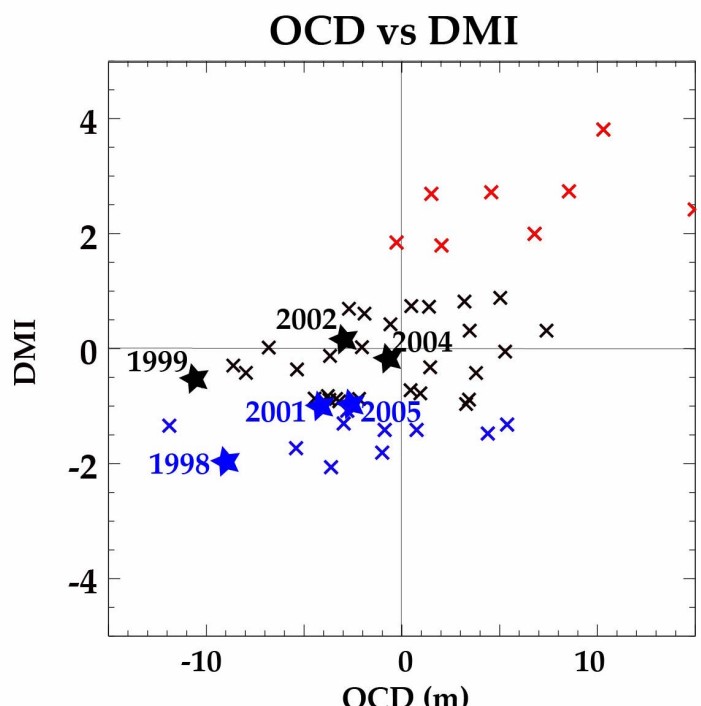

**Figure 12.** Scatterplot of the fall interannual anomalies of modeled WCI oxycline depth against modeled DMI. Red and blue crosses respectively indicate positive and negative IOD events (defined as events when DMI exceed one standard deviation). Years of anoxic events reported by Naqvi et al. (2009) are marked as stars along with the corresponding years.

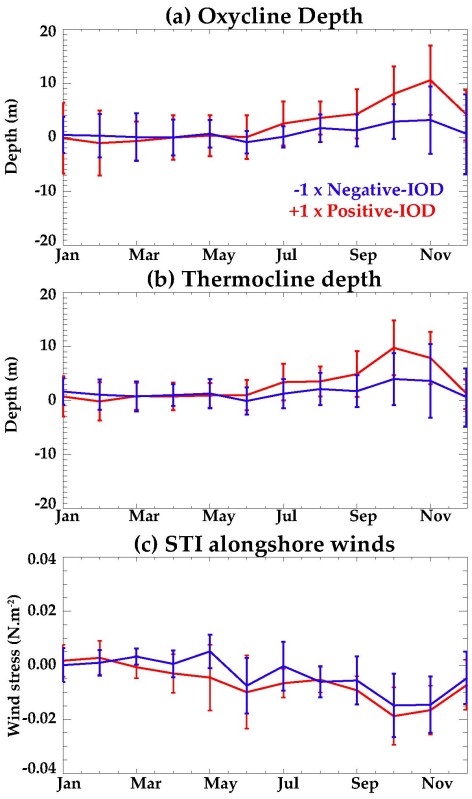

**Figure 13.** Seasonal evolution of anomalous composites of **(a)** WCI OCD, **(b)** WCI TCD and **(c)** STI zonal wind stress during positive (red) and negative (blue) IOD events. Positive (negative) IODs are defined when the DMI averaged for fall season is greater (lesser) than 1° C. The whiskers indicate the 95% confidence interval on the composited value. Positive IOD events considered in this composite are years 1961, 1963, 1967, 1972, 1977, 1982, 1994, 1997, and 2006 while negative ones are 1964, 1973, 1974, 1975, 1979, 1981, 1984, 1992, 1996, 1998, and 2010. Negative IOD composite time series has been multiplied by -1 to ease comparison with positive events.

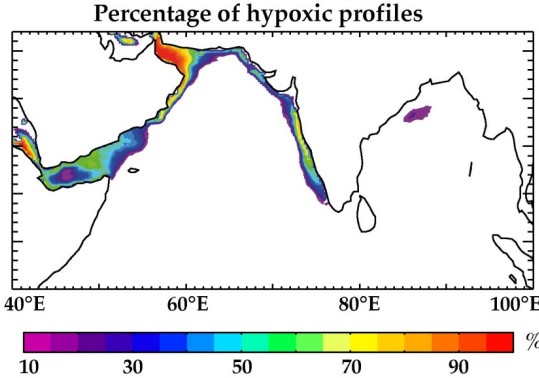

**Figure 14.** Percentage of occurrence of near-surface hypoxic conditions in the model during fall. Near-surface hypoxic conditions are defined as profiles with oxygen concentration below 80 $\mu$mol.l$^{-1}$ (a threshold under which a most large marine organism suffer from stress) in the upper 50 m.