# Peer review of "Positive Indian Ocean Dipole events prevent anoxia off the west coast of India"

_Biogeosciences, 2016_

## Referee Comment (RC1) · Anonymous Referee #1 · 26 Aug 2016

The paper by Parvathi et al. investigates the seasonal and interannual variability of dissolved oxygen off the West Coast of India (WCI) using a coupled physicalbiogeochemical regional simulation of the North Indian Ocean. More specifically, the study documents the recent variability in the oxycline along the WCI and explores its potential drivers. It is shown that the seasonal as well as the year-to-year fluctuations in the depth of hypoxia (and hence the likelihood of occurrence of anoxic events in the near-surface coastal waters) are controlled to a large extent by the variability of the depth of the thermocline, and hence are essentially physically driven. Authors show that oxycline like thermocline is shallowest in fall, thus eventually allowing anoxia to develop and reach the coastal region during this period of the year. Finally, it is shown that the year-to-year fluctuations in the oxycline (and hence the potential severity of hypoxia) in the WCI region is partially controlled by the Indian Ocean Dipole (IOD). This

is because of the sensitivity of the thermocline in this region to coastal Kelvin waves generated by easterly wind anomalies associated with IOD. Yet this effect is only strong during positive IODs where the downwelling Kelvin waves generated by easterlies near the southern tip of India lead to a deepening of the oxycline that prevents anoxia. In contrast, during negative IODs the effect is of smaller amplitude and hence is less important. The authors conclude that these findings have important implications in terms of the predictability of fall anoxia along WCI.

General comment:

The subject of the paper is highly relevant in the general context of quantifying and better understanding the drivers of naturally occurring coastal anoxia and what controls its year-to-year variability. The paper is well written and presents important new findings. The experimental design is appropriate despite some limitations (e.g., model resolution). For these reasons I recommend the publication of this manuscript after improving the presentation and the discussion of certain aspects of the study, namely the potential role of biology in contributing to O2 variability and the implications of the model low-resolution for the conclusions of the study. Here below I detail my comments:

Specific comments:

page 2, line 16: "The frequent anoxic conditions that occurred..."

page 4, line 13: The 1/4 horizontal resolution does not allow a proper representation of coastal upwelling either. The underestimation or potential misrepresentation of coastal (wind-driven) upwelling in the model needs to be mentioned and its potential implications discussed.

page 5, lines 9-20: in addition to the limitations discussed by authors, the CaTS dataset comes from a single coastal site and hence may not be representative of the dynamics of the whole WCI. This needs to be stressed.

page 6, 2nd equation: all parameters (R1o:c, R2o:c, etc,...) should be explicitly defined

BGD
in the text.

page 6, Is sediment respiration represented in the model? Please specify. Also, how are denitrification and anamox represented in the model?

Page 9, lines 14-15: I don't think the effect of biology on oxygen is small at the thermocline depth. The respiration fluxes can be very large at 100m. The fact that the oxycline and the thermocline show high correlations may result from the fact that biology itself is constrained by vertical physics and as a consequence that the nutricline and the thermocline are tightly coupled. This statement needs to be reformulated and the role of biology further discussed (authors may consider quantifying the individual contributions of biology and physical transport to oxygen variability).

Page 9, lines 18-22: It is not clear why the correlation between the OCD and the TCD is low in the southern and the southwestern parts of the domain. In the southwestern part authors mention a possible role of lateral advection there, but it is not clear how this might affect the correlation between these two quantities. In the equatorial region authors suggest the low correlation may be due to the definitions used for oxycline and thermocline that may not be meaningful there. Yet, in section 2.4 (page 8, lines 28-31) it is stated that the results and the conclusions of the study are insensitive to the oxygen (50-150 mmol/m3) and temperature (20-25C) criteria used to define these two depths. These two statements appear to contradict each other.

Page 9, lines 25-27: during the spring inter-monsoon, the (northwesterly) winds along the WCI are also upwelling-favorable. Yet, the thermocline is relatively deep there (especially in observations). This is an indication that TCD alone cannot be used as a proxy for wind-driven coastal upwelling (probably because of the non-local effects of the coastal Kelvin waves).

Page 10, lines 4-5: in the northern part of the WCI, alongshore (southward) winds seem to be upwelling-favorable. Maybe plotting seasonal time series of upwelling index together with OCD and TCD would help figuring out how these three quantities co-vary
(can be added to Fig 7).

Page 10, lines 24-32: the overestimation of oxygen in the WCI may not only be due to the coarse representation of the shelf dynamics but also the general tendency of the model to underestimate the intensity of the OMZ in the northern Arabian Sea (see Fig 4 for example). This needs to be discussed.

Page 13, lines 7-8: Identifying the drivers of coastal anoxia requires further investigation. In particular, the role of biology and the effects of O2 consumption on the shelf need to be quantified and contrasted with the impact of vertical and lateral advection. It is not clear how much of anoxia is driven by large-scale advection of O2-depleted water (from the Arabian Sea OMZ) vs. local consumption of O2 due to respiration in the water column and on the shelf.

Page 13, lines 17-20: Any explanation for the asymmetry of the impacts of positive and negative IODs (and in particular the potential reason for why the wind anomalies are weaker during negative IODs)?

Page 15, lines 11-14: As acknowledged by the authors, the (relatively) coarse resolution of the model (1/4) is probably the main limitation of the study. The underestimated or misrepresented coastal upwelling and mesoscale eddies in the model may have important consequences. I encourage the authors to further discuss the implications this might have and how it may (or may not) affect their conclusions/results.

Page 15, lines 11-16: if the model does not represent sediment processes, the lack of benthic respiration could also limit the model ability to represent the dynamics of coastal hypoxia along the shelf. This needs to be clarified.

Figure 3: the colorbar is missing.

Figure 3: since the study focus is on coastal anoxia that develops during fall, maybe it would be good to show how the model reproduces chlorophyll during this season.

---

## Referee Comment (RC2) · Anonymous Referee #2 · 23 Sep 2016

General comments (shared)

Ultimately, I think that there could be a linkage whereby positive IOD occurrence influences OCD conditions along the SW coast of India. The authors have taken a great stab at establishing this link but have fallen a bit short, in my opinion, of establishing the case. Significantly tightening up the analysis, following the suggestions I have made in specific comments below, would go a long way toward achieving a reasoned, well considered analysis that can at least strongly suggest such a linkage is present. It may be that additional data, refinement of this analysis in terms of key locations for exploring causality and higher resolution modeling with nested shelf regions, will be required to fully reveal what is tantalizingly indicated in the current effort.

Specific comments

[Figure]

Introduction, p. 2 & 3. The bottom paragraph of p. 2 and top paragraph of p. 3 have diverted from providing background information of the research to be undertaken to an initial summarizing, drawn from figure 1, of aspects of the study that is being reported on. This leads to an awkward shifting between presenting problem framework and results that interferes with coherent, logically developed reporting. I would recommend that all call outs to Fig. 1 be eliminated from the Introduction section. This could be simply accomplished by shifting the two noted paragraphs out, but note that there is at least one additional callout to Fig. 1 to address. As to the content of these two paragraphs, I do have additional specific comments. In the first of these paragraphs, the authors first introduce the notion that the open ocean OMZ of the Arabian Sea has connectivity to the hypoxia that manifests along the west coast of India. From my knowledge of the literature, and from the literature the authors' have cited, this has so far only been posited and in my view is more anecdotal than proven fact. The $1/4°$ model solution that the authors use as principal basis of their findings is challenged to demonstrate causality on this point. The authors should acknowledge this explicitly, make a more compelling case based on existing literature, or leverage this in from some of their higher resolution modeling efforts.

Page 5, line 20. In the last sentence, it is interesting to note that these observed instances of full anoxia will be featured/discussed later in the paper but the figure callout should not be included. Could probably accomplish this through merging of that point into the prior sentence and eliminating the remainder.

Page 6, line 8. In this equation describing the biological source / sink terms of DO, it would be interesting to note what is done to ensure negative concentrations of dissolved oxygen are not achieved in the model.

Page 9, lines 1-2. The text here is a bit confusing. At the top of the paragraph it is stated: ". . . we used the standard Dipole Mode Index . . . ". From the subsequent text in this section that indicates DMI based on model SST is calculated, I think what is meant is " . . . the standard DEFINITION of Dipole Mode Index . . . " is used.

[Figure]

Page 9, line 6. " to the choice of either of them" is awkward. "which is chosen" would work better.

Page 9, line 26 -> top page 10. In the model during MAM, Fig. 6e shows that OCD is not really uniform in eastern AS; and OCD off Mumbai is comparable in value to OCD at bottom of the subcontinent. This directly contrasts what is described in the text. It also dampens the scenario described as a clear-cut influence of coastal KW propagation northward of the shoaled OCD / TCD condition that initiates in the STI region. I believe that the issue here is that care needs to be taken to make clear that description is centered on WCI to STI region and not relevant north of $\sim 10°$ where alongshore wind forcing is distinct from what exists farther south.

Page 10, line 8. I think it is stretching what can be elucidated from the model results and presented figures to say that the OCD/TCD pattern is clearly suggestive of faster planetary wave propagation at lower latitudes. Lag associated with coastal KW propagating northward could also factor in.

Page 10, bottom paragraph. The linkage of offshore OMZ to low coastal DO is again a thread in this part of text (see earlier remarks on my reservations).

Page 11, lines 3 - 17, and figure 9. There are a number of interesting features in figure 9. Some comment on the inverse correlation regions appearing in 9a would be interesting to include and may provide some mechanistic insight to IOD-associated biophysical interaction that is the crux of this analysis. Some comment on the several instances where model TCD and observed SLA are out of phase in the 2002-2006 period (9b) would also be potentially illuminating (model issue?, sensitivity to WCI box definition?). Assessing any limitations in either of these is key to explore and characterize for the reader to fully trust results stemming from this analysis.

Page 11, lines 19-20. Details of how anomalies are determined would be useful to document; caption of Fig. 10 is an option if this is not substantial enough to stand alone in methods section. Also, on line 20 panels a-c are noted as regression maps, which

is contradictory to the information in the Fig. 10 caption (and plot labeling) that states panel c is a correlation map. For both of these data reductions details of how they are performed are not documented, making it problematic for the reader to correctly self-determine his/her interpretation. As for grasping why distinct analysis method was applied to zonal wind stress relative to how the other variables (OCD, TCD, SST) were treated, that is even more of a challenge for the reader to intuit.

Page 11, lines 19 - 23 and caption for Fig. 10. The terminology used in referring to the derived fields described here, and the terminology in the captions that accompany the associated distributions in Fig. 10, is inconsistent. Specifically, the narrative notes that all variables are internally varying anomalies but in the caption this is not obvious. In the caption, the data shown in panels a-c are noted to be regressed against "normalized oxycline interannual anomalies averaged over WCI box". My interpretation is that this describes the ts shown in Fig. 11a. If this is indeed the case, noting that as such would be very helpful to the reader as a way to better grasp what is presented in Fig. 10. This may entail modifying figure ordering w/in the ms. I find it interesting that the regression of OCD with OCD(WCI) in the WCI box (panel 10a) is actually relatively low compared to elsewhere in the IO domain. I think some interpretive commentary from the authors would be very interesting to see. Further, as I questioned earlier, does this have implications for the sensitivity / utility / robustness of the WCI box as a foundational component of this analysis? I would like to see the authors critically assess and comment on the choices they have made in setting up and carrying out their analysis.

Page 11, line 24. For clarity and to benefit the reader, please explicitly associate / define how shallower / deeper OCD anomaly relates to + / - OCD anomaly.

Page 11, lines 25-31. The KW propagation patterns noted here are consistent with what is reported in the literature. However, the pathways and timings that are discussed are not identifiable in the Fig. 10. Appropriate referencing of the literature should be given to support what is stated. The remote forcing aspect of thermocline dynamics in

the northern IO is quite complex; I strongly encourage the authors to make the effort to clearly articulate and document what is known and how it relates to their study. I believe this would be highly appreciated by the IO readership.

Page 11, line 31. Should specify that the correlation referred to here is negative.

Page 12, lines 4-8. The correlation distribution pattern for taux (Fig. 10c) is different enough that I would hesitate to even characterize it as "reminiscent of IOD signature"; in particular the sign shift in correlation that is apparent in the NE Bay of Bengal and eastward / off equator shift of strongest correlation (i.e., away from Sumatra coast where + IOD signature is most pronounced). Note as well that this is positive IOD signature. And related to that call for clarification, it would be nice to include an example of negative IOD manifestation to accompany the example of positive IOD (Fig. 5d).

Page 12, line 6. Here, the 10 d-f sequence is collectively referred to as regression maps, which is inconsistent with labeling / reporting elsewhere in narrative.

Page 12, lines 8-9. I think it is a stretch to make the statement that these figures, in particular panel 10d which is the key for illustrating the point, demonstrate a "strong link" between OCD in WCI region and IOD. In my view the level of regression in 10d for the WCI region is marginal and certainly much less pronounced than elsewhere (e.g., the Sri Lanka Dome).

Page 12, lines 10-11. The correlation between model-based DMI and OCD(WCI) is interesting and suggestive of what the authors are trying to demonstrate (i.e., causal link between OCD and DMI). I would strongly suggest that this correlation also be performed with standard DMI product, for a couple of reasons. While the authors did report earlier in the ms that the standard DMI and their model-based DMI were largely similar and choice of which is applied did not substantially affect the results, performing the correlation with standard DMI provides grounding to a well-known established index and would mitigate any concerns that may/should arise for the reader vis a vis internal bias inherent in a model-model comparison.

Page 12, lines 14-21, and Fig. 11b. This is an interesting distillation of results, though I still have reservations of robustness similar and related to what has been noted in comments above. I think it would be useful to find a way to split out the DMI values such that those that pass threshold and can be classified as positive / negative IOD are distinct from those that do not (i.e., normal and IOD states are clearly delineated). The regressions that are shown do not add much insight, but if they are retained then they should be performed only on points that are +/- IOD states and not inclusive of all positive of negative values. However, even doing this, I am skeptical that particularly insightful result will be obtained. For the data that are shown that do represent + / - IOD occurrence, there is not really an associated systematic deepening / shoaling of OCD. Certainly the negative IOD case has a number of either OCD result. But there is also a positive IOD case with (slightly) shallowing OCD. It also appears that the years with observed and notable anoxic events tend to be during negative DMI (thought not necessarily negative IOD) but there is also a positive DMI case and a case that is almost uniformly normal (i.e., DMI $\sim$ 0 and OCD $\sim$ 0). Given these results, I do consider that the data are suggestive of positive IOD events influencing appearance of low oxygen in WCI but would be hesitant to make categorical conclusions. Overall, I think the authors need to be more even handed when reporting what is revealed in Fig. 11b.

Page 12, lines 21-27, and Fig. 12. This figure is also quite interesting. It suggests that both IOD phases lead to positive OCD and TCD displacement during SON, although for negative IOD this may not be statistically significant. What I find curious is that taux during SON in STI that has opposite sign between positive and negative IOD. Would this not lead to opposite oceanic response (i.e., upwelling vs. downwelling) between the two IOD phases, which then presumably has contrasting impact on thermocline displacement if this response translates northwards as coastal KW as authors have argued? The authors do note that taux condition for negative IOD is not as "robust" as that for positive IOD, however the relative values of anomaly for October (highest magnitude for SON period) are not strikingly distinct. Which begs the question as to

how much OCD / TCD response one can ascribe to STI wind stress condition.

Page 12, slide 30. Instead of "identified fifteen years ago", reiterating citation of relevant source material would be best.

Page 20, line 8. Title for this reference is incorrect.

Comments on Figures

General comment. There are several figures that include isolines superimposed on a second variable where a color scale is applied. In almost all cases, there is a need for additional labeling of the superimposed contour lines on the plot so these distributions can be grasped. There is also a need to note the contour intervals in the associated captions.

Figure 4. For panels 4b and 4d, the red contour line demarking thermocline is extremely difficult to see. In addition, the isotherms at depth (DO < 80 microM) are also difficult to see. For panels 4a and 4c, the 15° N line is difficult to spot as well.

Figure 9. In panel 9a, white color in the IO domain has non-unique meaning. It can be either masked data or the transition from positive to negative correlation value. A way of uniquely distinguishing these should be used. Additionally, the white land mask is also not ideal but does have benefit of land-sea boundary.

Panels 11a and 12 a-c. The combination of blue and black line colors is ineffective. Very difficult to distinguish between these two.

END OF REVIEW

---

## Author Comment (AC1) · 28 Oct 2016

*The referee comments are italicized.* Answers in regular typeface. Actions taken in red.

**Answers to referee 1 comments**

*Anonymous Referee #1:*
*Summary: The paper by Parvathi et al. investigates the seasonal and interannual variability of dissolved oxygen off the West Coast of India (WCI) using a coupled physical-biogeochemical regional simulation of the North Indian Ocean. More specifically, the study documents the recent variability in the oxycline along the WCI and explores its potential drivers. It is shown that the seasonal as well as the year-to-year fluctuations in the depth of hypoxia (and hence the likelihood of occurrence of anoxic events in the near-surface coastal waters) are controlled to a large extent by the variability of the depth of the thermocline, and hence are essentially physically driven. Authors show that oxycline like thermocline is shallowest in fall, thus eventually allowing anoxia to develop and reach the coastal region during this period of the year. Finally, it is shown that the year-to-year fluctuations in the oxycline (and hence the potential severity of hypoxia) in the WCI region is partially controlled by the Indian Ocean Dipole (IOD). This is because of the sensitivity of the thermocline in this region to coastal Kelvin waves generated by easterly wind anomalies associated with IOD. Yet this effect is only strong during positive IODs where the downwelling Kelvin waves generated by easterlies near the southern tip of India lead to a deepening of the oxycline that prevents anoxia. In contrast, during negative IODs the effect is of smaller amplitude and hence is less important. The authors conclude that these findings have important implications in terms of the predictability of fall anoxia along WCI.*

*General comment:*
*The subject of the paper is highly relevant in the general context of quantifying and better understanding the drivers of naturally occurring coastal anoxia and what controls its year-to-year variability. The paper is well written and presents important new findings. The experimental design is appropriate despite some limitations (e.g., model resolution). For these reasons I recommend the publication of this manuscript after improving the presentation and the discussion of certain aspects of the study, namely the potential role of biology in contributing to O2 variability and the implications of the model low-resolution for the conclusions of the study.*

We thank the referee for his encouraging review. As detailed below, we propose to improve the presentation and discussion of the potential role of biology in contributing to O2 variability and the implication of the low-resolution of the model used. The updated figures that we will provide in the revised version are also displayed at the end of this document.

*Here below I detail my comments:*
*Specific comments:*
*Page 2, line 16: "The frequent anoxic conditions that occurred…"*

Corrected.

*Page 4, line 13: The 1/4 horizontal resolution does not allow a proper representation of coastal upwelling either. The underestimation or potential misrepresentation of coastal (wind-driven) upwelling in the model needs to be mentioned and its potential implications discussed.*

We briefly mention this model limitation in the introduction section but defer to the discussion section for more extensive discussion regarding the implications of the underestimation of coastal upwelling. The discussion on the model limitation will be expanded along the following lines: "The spatial resolution of our regional model (1/4°) is sufficient to describe the offshore oxygen variability, but probably insufficient to properly resolve the dominant physical processes controlling the upwelling dynamics (e.g. Huthnance 1995; Allen et al. 2010). For the case of the narrow continental shelf along the west coast of North America, several studies have shown that a minimum horizontal resolution of 10 km is indeed required (Marchesiello et al., 2003; Veneziani et al., 2009). Our current model resolution is not sufficient to properly resolve the oceanic mesoscale eddies that have been shown to

be important for the exchanges between the shelf and the open ocean in upwelling regions (e.g. Marchesiello et al. 2003 ; Bettencourt et al. 2015 ; Vergara et al. 2016). Another limitation arises from the absence of tidal forcing in the model, which may play an important role, as strong internal tides can be generated on the shelf break and contribute to enhance the thermocline vertical excursion and mixing (Monteiro et al. 2005). Despite these limitations, the model presented in this study provide some evidence for a control of the seasonal and interannual oxygen concentration on the west Indian shelf by offshore oxygen fluctuations, as suggested in the seminal studies of Banse (1959) and Carruthers et al. (1959). In order to establish an unequivocal evidence for the shelf-open ocean interactions, future studies should consider increased model resolution in the shelf region and improved observations such as repeated glider transects or triad of moorings (shelf, shelf break and open ocean) in this region."

Related references :

Allen, J. I., Aiken, J., Anderson, T. R., Buitenhuis, E., Cornell, S., Geider, R. J., ... & Hardman-Mountford, N. : Marine ecosystem models for earth systems applications: The MarQUEST experience. Journal of Marine Systems, 81(1), 19-33, 2010.

Banse, K.: On upwelling and bottom-trawling off the south west coast of India, J. Mar. Biol. Assoc. India, 1, 33–49, 1959.

Bettencourt JH, López C, Hernández-García E, et al: Boundaries of the Peruvian oxygen minimum zone shaped by coherent mesoscale dynamics. Nature Geoscience 8:937–940. doi: 10.1038/ngeo2570, 2015.

Carruthers, J. N., Gogate, S. S., Naidu, J. R., and Laevastu, T.: Shoreward upslope of the layer of minimum oxygen off Bombay: Its influence on marine biology, especially fisheries, Nature, 183,1084–1087, 1959.

John M. Huthnance, Circulation, exchange and water masses at the ocean margin: the role of physical processes at the shelf edge, Progress in Oceanography, Volume 35, Issue 4 Pages 353-431, ISSN 0079-6611, http://dx.doi.org/10.1016/0079-6611(95)80003-C, 1995.

Marchesiello, P., McWilliams J.C., Shchepetkin, A.F. :Equilibrium structure and dynamics of the California Current System. J Phys Oceanogr 33:753–783, 2003.

Monteiro, P.M.S., Nelson, G., van der Plas, A., Mabille, E., Bailey, G.W., Klingelhoeffer, E.: Internal tide-shelf topography interactions as a potential forcing factor govern- ing the large scale sedimentation and burial fluxes of particulate organic matter (POM) in the Benguela upwelling system, Continental Shelf Research, 25, 1864–1876, 2005.

Veneziani, M., Edwards, C.A., and Moore, A.M.: A central California coastal ocean modeling study: 2. Adjoint sensitivities to local and remote forcing mechanisms, J. Geophys. Res., 114, C04020, doi:10.1029/2008JC004775, 2009.

Vergara, O., Dewitte, B., Montes, I., Garçon, V., Ramos, M., Paulmier, A., and Pizarro, O.: Seasonal variability of the oxygen minimum zone off Peru in a high-resolution regional coupled model, Biogeosciences, 13, 4389-4410, doi:10.5194/bg-13-4389-2016, 2016.

*Page 5, lines 9-20: in addition to the limitations discussed by authors, the CaTS dataset comes from a single coastal site and hence may not be representative of the dynamics of the whole WCI. This needs to be stressed.*

Thank you for pointing this out. This will be mentioned in the revised draft as follows: "In addition, these observations collected at single coastal site may be influenced by local processes and hence may not be representative of the dynamics over the entire WCI."

*Page 6, 2nd equation: all parameters ($R^1_{o:c}$, $R^2_{o:c}$, etc…) should be explicitly defined in the text.*

All parameters are now explicitly defined in the revised draft.

*Page 6, Is sediment respiration represented in the model? Please specify. Also, how are denitrification and anamox represented in the model?*

A more thorough discussion of these modelling aspects will be included in the model description section of the paper along the following lines: "When oxygen falls below a threshold value, which is

set to 6 µM, nitrate instead of oxygen starts to be increasingly consumed during the remineralization of the organic matter, i.e., denitrification. Anamox is not represented in the model. At the bottom of the ocean, the model includes a very simple sedimentation process. The metamodel of Middelburg et al. (1996) is used to compute the relative contribution of the denitrification to the remineralization of the organic matter. The flux of organic matter to the sediment is then used to compute the oxygen and nitrate demands in the sediment, which are then imposed as boundary conditions to the model."

Related reference:

Middelburg, J. J., Soetaert, K., Herman, P. M., & Heip, C. H.: Denitrification in marine sediments: A model study. *Global Biogeochemical Cycles*, *10*(4), 661-673, 1996.

*Page 9, lines 14-15: I don't think the effect of biology on oxygen is small at the thermocline depth. The respiration fluxes can be very large at 100m. The fact that the oxycline and the thermocline show high correlations may result from the fact that biology itself is constrained by vertical physics and as a consequence that the nutricline and the thermocline are tightly coupled. This statement needs to be reformulated and the role of biology further discussed (authors may consider quantifying the individual contributions of biology and physical transport to oxygen variability).*

Resplandy et al. (2012) did already quantify the respective contributions of biology and vertical transport to the seasonal oxygen variability along the west coast of India. Despite the compensation of biological sink and dynamical source of oxygen on an annual average, they found that the seasonality in the dynamical transport of oxygen is 3 to 5 times larger than that in the biological sink. The seasonality in oxygen along the WCI hence primarily arises from the vertical displacement of the oxycline attributed to the influence of northward propagating coastal Kelvin waves forced remotely from the Bay of Bengal. This conclusion does not support the hypothesis of Sarma (2002), who suggested that the weak oxygen seasonality could arise from compensation between the dynamical input and the biological uptake of oxygen during each season. This hypothesis could not however be verified owing to the absence of reliable seasonal estimates of oxygen consumption (Sarma, 2002). The text will be modified to explicitly mention the dominance of vertical transport on the modelled seasonal oxygen variations in this region derived from the oxygen budget analysis of Resplandy et al. (2012).

Related references:

Resplandy, L., Lévy, M., Bopp, L., Echevin, V., Pous, S., Sarma, V. V. S. S., and Kumar, D.: Controlling factors of the oxygen balance in the Arabian Sea's OMZ, Biogeosciences, 9, 5095–5109, doi:10.5194/bg-9-5095-2012, 2012.
Sarma, V. V. S. S.: An evaluation of physical and biogeochemical processes regulating perennial suboxic conditions in the water column of the Arabian Sea, Global Biogeochem. Cy., 16, 1082, doi:10.1029/2001GB001461, 2002.

*Page 9, lines 18-22: It is not clear why the correlation between the OCD and the TCD is low in the southern and the southwestern parts of the domain. In the southwestern part authors mention a possible role of lateral advection there, but it is not clear how this might affect the correlation between these two quantities.*

If any other process than vertical advection strongly contributes to OCD variability (i.e. lateral advection, biology, etc…), there is no reason for OCD variability to be in phase with TCD variability and correlation between the two variables should hence decrease.

*In the equatorial region authors suggest the low correlation may be due to the definitions used for oxycline and thermocline that may not be meaningful there. Yet, in section 2.4 (page 8, lines 28-31) it is stated that the results and the conclusions of the study are insensitive to the oxygen (50-150 mmol/m3) and temperature (20-25C) criteria used to define these two depths. These two statements appear to contradict each other.*

We agree that these two statements are misleading. We meant that main results regarding the variability of the OCD/TCD variations in the Arabian Sea and more specifically in our focus region along the WCI are not sensitive to that choice. This will be clarified in the revised manuscript.

*Page 9, lines 25-27: during the spring inter-monsoon, the (northwesterly) winds along the WCI are also upwelling-favorable. Yet, the thermocline is relatively deep there (especially in observations). This is an indication that TCD alone cannot be used as a proxy for wind-driven coastal upwelling (probably because of the non-local effects of the coastal Kelvin waves).*

We agree with the reviewer. This will be mentioned in the revised manuscript as follows: "During the spring inter monsoon (March-May), the TCD and OCD are spatially quite uniform and deep (~100 m) in the southeastern Arabian Sea (Figs. 5a and 5e). This is also the case along the WCI south of 15°N despite the local upwelling favorable alongshore winds, indicating a remote control of OCD and TCD variations there."

*Page 10, lines 4-5: in the northern part of the WCI, alongshore (southward) winds seem to be upwelling-favourable. Maybe plotting seasonal time series of upwelling index together with OCD and TCD would help figuring out how these three quantities co-vary (can be added to Fig 7).*

The paper will be revised to include a more thorough discussion of the local and remote wind control on the OCD and TCD variations along the west coast of India. The updated Figure 7 now provides a direct comparison between the wind seasonal variations along the coast and at the southern tip of India and the west coast of India temperature and oxygen seasonal variability in both the model and observations. This figure shows that the TCD starts to shallow in April and becomes shallowest in September-October, the peak season of the upwelling. Local alongshore winds along the WCI are favourable to upwelling only during the southwest monsoon, indicating the influence of remote wind forcing on the upwelling along the WCI.

*Page 10, lines 24-32: the overestimation of oxygen in the WCI may not only be due to the coarse representation of the shelf dynamics but also the general tendency of the model to underestimate the intensity of the OMZ in the northern Arabian Sea (see Fig 4 for example). This needs to be discussed.*

We don't agree with that statement. The good agreement between the model and WOA seasonal oxygen variations in the WCI box (Fig. 7) suggests that the model is able to accurately capture the oxygen variations offshore the WCI. This good agreement suggests that the model OMZ underestimation in the northwestern AS (Fig. 3ac) does not affect the oxygen representation along the WCI. It is hence more likely that shelf-specific processes may explain the lower oxygen contents observed at CaTS as compared to the model and WOA data.

*Page 13, lines 7-8: Identifying the drivers of coastal anoxia requires further investigation. In particular, the role of biology and the effects of O2 consumption on the shelf need to be quantified and contrasted with the impact of vertical and lateral advection. It is not clear how much of anoxia is driven by large-scale advection of O2-depleted water (from the Arabian Sea OMZ) vs. local consumption of O2 due to respiration in the water column and on the shelf.*

We agree with the reviewer that this statement is too strong, given our model limitations. We tuned down that statement in the revised manuscript. In addition, a revised discussion on the need of a refined assessment of the large-scale and local biological processes influencing anoxia along the shelf will be provided in the discussion section along the following lines: "In this paper, we used the model and WOA data as representative of the open ocean behavior, offshore off the shelf break along the west coast of India. Several studies have already pointed towards the influence of offshore oxygen variations in driving the variability of hypoxic conditions along other coastal regions (e.g. Grantham et al. 2004; Helly and Levine, 2004; Arntz et al., 2006; Gutierrez et al., 2008). As was shown on figures 6 and 7b, the model and WOA climatology vertical oxygen distribution agree quite well, both in terms

of the oxycline depth and near-surface value. The CaTS data, on the other hand, is representative of what happens much closer to the coast and displays much lower oxygen values than seen further offshore in WOA and the model. This may of course partially be due to shortcomings in representing physical exchanges between the shelf and open ocean at the current resolution of our model and existing oxygen dataset in the region. But biological processes are also known to be a prominent oxygen consumption term on the shelf, in particular in the benthic zone where the enhanced concentration of particulate matter above sediments is associated with high oxygen demand (e.g. Cowie, 2005). The crude parametrization of sediments in the model probably does not consume enough oxygen very close to the coast. On the other hand, the good phasing between the O2 seasonal variability offshore (in the model and WOA) and CaTS shelf data (Fig. 7bc) suggests that the offshore variability is probably an important driver of the O2 content on the shelf. A proper representation of benthic biological processes would however probably be needed to represent the low O2 absolute values very close to the coast (Fig. 7c). It is also likely that the anthropogenic and natural fertilizing inputs along coast are not well reproduced in the model, and that the resulting underestimated near-shore biological productivity also contributes to an underestimated oxygen demand. Dedicated studies at higher spatial resolution with sensitivity tests on the representation of near-shore biological processes will probably be needed in order to better understand how the representation of near-shore biological processes constrain the coastal O2 representation. Observed cross-shore sections of O2 and physical properties will also be necessary to be able to validate the model, which is difficult with the currently available datasets in this region."

Related references:

Arntz WE, Gallardo VA, Gutiérrez D, et al (2006) El Niño and similar perturbation effects on the benthos of the Humboldt, California, and Benguela Current upwelling ecosystems. Advances in Geosciences 6:243–265. doi: 10.5194/adgeo-6-243-2006.

Cowie G.L., The biogeochemistry of Arabian Sea surficial sediments: a review of recent studies, Prog. Oceanogr., 65 (2005), pp. 260–289

Gutierrez, D., Enriquez, E., Purca, S., Quipuzcoa, L., Marquina, R., Flores, G., and Graco, M.: Oxygenation episodes on the continental shelf of central Peru: Remote forcing and benthic ecosystem response, Prog. Oceanogr., 79, 177–189, 2008.

Grantham BA, Chan F, Nielsen KJ, et al (2004) Upwelling-driven nearshore hypoxia signals ecosystem and oceanographic changes in the northeast Pacific. Nature 429:749–754. doi: 10.1038/nature02605.

Helly, J. J. and Levin, L. A.: Global distribution of naturally occurring marine hypoxia on continental margins, Deep-Sea Res. I, 51, 1159–1168, 2004.

Monteiro P, Van Der Plas AK, Melice JL (2008) Interannual hypoxia variability in a coastal upwelling system: Ocean–shelf exchange, climate and ecosystem-state implications. Deep Sea Research …. doi: 10.1016/j.dsr.2007.12.010

*Page 13, lines 17-20: Any explanation for the asymmetry of the impacts of positive and negative IODs (and in particular the potential reason for why the wind anomalies are weaker during negative IODs)?*

Negative IODs tend to be weaker than their positive counterpart due to nonlinear response of the deep atmospheric convection to SST anomalies (Saji and Yamagata 2003; Hong et al. 2008; Cai et al. 2013). The differences in wind patterns (the fact that for a given DMI absolute value, positive IODs tend to display wind anomalies that extend more northward) may also be related to this asymmetry. This will be mentioned in the revised manuscript as follows: "Our results suggest that the weaker WCI oxycline depth response during negative IODs may partly be explained by the weaker STI winds anomalies associated with negative IOD events. This weaker wind amplitude could be simply related to the tendency of negative IOD events to be weaker than their positive counterpart (Saji and Yamagata 2003; Hong et al. 2008; Cai et al. 2013) or to asymmetries in the wind spatial patterns associated with the non-linear response of deep atmospheric convection to SST anomalies of each phase of the IOD. A more precise understanding of this asymmetry would require an in-depth investigation of the processes that control the STI wind variations and the thermocline along the WCI in response to positive and negative IOD events."

*The referee comments are italicized.* Answers in regular typeface. Actions taken in red.

Related references:

Cai, W., Zheng, X. T., Weller, E., Collins, M., Cowan, T., Lengaigne, M., ... & Yamagata, T. (2013). Projected response of the Indian Ocean Dipole to greenhouse warming. *Nature Geoscience*, *6*(12), 999-1007.

Hong, C. C., Li, T., & Kug, J. S. (2008). Asymmetry of the Indian Ocean Dipole. Part I: Observational Analysis*. *Journal of climate*, *21*(18), 4834-4848.

Saji, N. H., & Yamagata, T. (2003). Possible impacts of Indian Ocean dipole mode events on global climate. *Climate Research*, *25*(2), 151-169.

*Page 15, lines 11-14: As acknowledged by the authors, the (relatively) coarse resolution of the model (1/4) is probably the main limitation of the study. The underestimated or misrepresented coastal upwelling and mesoscale eddies in the model may have important consequences. I encourage the authors to further discuss the implications this might have and how it may (or may not) affect their conclusions/results*

This will be done in the revised discussion (cf answer to your first comment).

*Page 15, lines 11-16: if the model does not represent sediment processes, the lack of benthic respiration could also limit the model ability to represent the dynamics of coastal hypoxia along the shelf. This needs to be clarified.*

The model does include a simple representation of sediment processes. The simplicity of this representation may however be a limit of the model's ability to represent the dynamics of coastal hypoxia. This will be clearly stated in the updated discussion of the potential role of biology that is quoted in answer to your comment above.

*Figure 3: the colorbar is missing.*

Sorry for this. Thank you for pointing this out. The colorbar is now added.

*Figure 3: since the study focus is on coastal anoxia that develops during fall, maybe it would be good to show how the model reproduces chlorophyll during this season.*

Figure R1 below provides the summer and fall Chl patterns for the model and observations. As shown in this Figure, the patterns during these two seasons share a lot in common, with a tendency for weaker Chl values in fall compared to summer in both model and observations. We hence did not include the fall Chl pattern in the revised manuscript but will clearly state that the fall Chl patterns are similar, although weaker, than those in summer in both observations and model.

*The referee comments are italicized.* Answers in regular typeface. Actions taken in red.

[Figure]

**Figure R1:** Northern Indian Ocean surface chlorophyll (mg.m$^{-3}$) climatology for (left) summer (June-August) and (right) fall (September-October) in (top) the ESA satellite product and (bottom) model.

*The referee comments are italicized.* Answers in regular typeface. Actions taken in red.

**Answers to referee 2 comments**

*General comments : Ultimately, I think that there could be a linkage whereby positive IOD occurrence influences OCD conditions along the SW coast of India. The authors have taken a great stab at establishing this link but have fallen a bit short, in my opinion, of establishing the case. Significantly tightening up the analysis, following the suggestions I have made in specific comments below, would go a long way toward achieving a reasoned, well considered analysis that can at least strongly suggest such a linkage is present. It may be that additional data, refinement of this analysis in terms of key locations for exploring causality and higher resolution modeling with nested shelf regions, will be required to fully reveal what is tantalizingly indicated in the current effort.*

We thank the reviewer for his encouraging comments. We followed most of the reviewer's suggestions, which led to significant changes in the Figures list as summarized below:

1- As suggested by the reviewer, results from Figure 1 have been integrated in the new Figures 6 and 7. The text related to the earlier figure has hence been dropped from the introduction and a more precise discussion on the past literature discussing the offshore/shelf oxygen connection is now included in the introduction.

2- New Figure 6 provides a comparison between O2 and temperature seasonal variations in the WCI box for both model and WOA data.

3- The updated Figure 7 gathers results from old Figure 7 augmented with an assessment of seasonal temperature variations along the shelf from CaTS measurements and local and remote (from the southern tip of India) seasonal wind variations. This updated Figure allows us discussing in more details: (1) the modelled and observed seasonal phasing between upwelling and oxygen depletion, pointing towards a strong physical control of seasonal oxygen variations, (2) the seasonal phasing between offshore and coastal temperature and oxygen variations, further highlighting the offshore/coastal connection at seasonal timescales and (3) the remote influence of STI wind seasonal variations on WCI upwelling variability.

4- The updated Figure 8 now includes timeseries of interannual oxycline and thermocline variations in the WCI box, further highlighting their in-phase relationship and hence the strong physical control onto WCI interannual oxycline variability.

5- The updated Figure 9 allows demonstrating the strong control of STI wind variability on observed and modelled WCI interannual thermocline variations through coastal Kelvin waves, along with the ability of the model to capture observed interannual thermocline variations in the WCI region.

6- The updated Figure 10 is a simplified version of previous Figure 10, from where the correlation between winds and the WCI oxycline / DMI index has been removed.

7- The updated Figure 11 demonstrates that (1) WCI interannual OCD variations are well correlated to the DMI index computed from either the model and observations and (2) that these variations are remotely controlled by interannual wind variations in the STI region rather than by local winds.

These new figures are appended below for your reference and the text will be changed accordingly. Below, we further provide point-by-point answer to each of your specific concerns.

*The referee comments are italicized.* Answers in regular typeface. Actions taken in red.

*Specific comments :*
*Introduction, p. 2 & 3. The bottom paragraph of p. 2 and top paragraph of p. 3 have diverted from providing background information of the research to be undertaken to an initial summarizing, drawn from figure 1, of aspects of the study that is being reported on. This leads to an awkward shifting between presenting problem framework and results that interferes with coherent, logically developed reporting. I would recommend that all call outs to Fig. 1 be eliminated from the Introduction section. This could be simply accomplished by shifting the two noted paragraphs out, but note that there is at least one additional callout to Fig. 1 to address. As to the content of these two paragraphs, I do have additional specific comments. In the first of these paragraphs, the authors first introduce the notion that the open ocean OMZ of the Arabian Sea has connectivity to the hypoxia that manifests along the west coast of India. From my knowledge of the literature, and from the literature the authors' have cited, this has so far only been posited and in my view is more anecdotal than proven fact. The 1/4_ model solution that the authors use as principal basis of their findings is challenged to demonstrate causality on this point. The authors should acknowledge this explicitly, make a more compelling case based on existing literature, or leverage this in from some of their higher resolution modelling efforts.*

The seminal paper by Banse (1959) shows a section across the shelf off Cochin located ~10$^{\circ}$N on the west coast of India. Based on the analysis of this section, Banse (1959) stated that the hypoxic conditions off Cochin were clearly connected with seasonal upwelling variations in relation with the southwest monsoon. Similar findings were reported off Bombay by Carruthers et al. (1959), who showed the layer of minimum oxygen was upsloping towards the coast right after the summer monsoon demise. Similar cross-shore sections along the west coast of India across the monsoon season are shown in Naqvi et al. (2009) and used to support the notion that it is the seasonal upwelling that brings to the west Indian continental shelf the oxygen-poor subsurface waters that then turn anoxic by late summer due to exhaustion of oxygen and nitrate by heterotrophic micro-organisms. Another recent paper (Gupta et al. 2016) further concluded that upwelling of oxygen-deficient waters during the monsoon is the major process regulating the biogeochemistry along this shelf using ten shelf transects during 2012 near 10°N. We believe that these observational analyses at different locations along the west coast of India are clear indications that there is some connection between open ocean seasonal oxygen variability and coastal hypoxia. We however agree with the reviewer that the discussion based on Figure 1 should rather belong to the result section. We also agree with the reviewer that the detailed processes by which these exchanges occur are not yet properly understood, nor resolved by our model or existing datasets. To address the reviewer's concerns, we will do the following changes:

1- All callouts to Figure 1 will be eliminated from the introduction. Results from this Figure will be incorporated in Figures 6 and 7 to discuss the connection between the offshore and costal seasonal O2 variability and the role of the remote and local seasonal alongshore wind variations in driving this variability in more detail.

*2-* Top paragraph of p3 will be removed from the introduction and will be discussed when analysing the updated Figures 6 and 7. We will rewrite the bottom paragraph of p2, to more precisely acknowledge on which observational basis the influence of upwelling has been reported as the major process driving the coastal hypoxia along the West Indian continental shelf. We will especially refer to the results of Banse et al. (1959, 1968), Carruthers et al. (1959), Naqvi et al. (2009) and Gupta et al. (2016) who all support this offshore-coastal connection based on the analysis of the cross-shore transects data at different locations along the west coast of India. We will however also clearly acknowledge in this paragraph that these conclusions were drawn from punctual cruises snapshots. Figure 7 provides a comparison of the seasonal oxygen and temperature evolution offshore and at the coast and of the remote and local alongshore winds. It allows clearly illustrating the strong seasonal phasing between the offshore and coastal evolution, further pointing towards a strong connection between open ocean OMZ and coastal hypoxia through the upwelling processes. This Figure also allows to demonstrate the remote control of the alongshore winds at the southern tip of India on the seasonal upwelling conditions along the west coast of India.

*The referee comments are italicized.* Answers in regular typeface. Actions taken in red.

3- We will clarify when introducing the model that it does not resolve the details of the shelf / open ocean interactions, but is merely used to understand the processes that control offshore variations. We will also include a discussion of the need for improved observations and modeling to resolve processes of shelf-open ocean exchanges in the discussion.

*Page 5, line 20. In the last sentence, it is interesting to note that these observed instances of full anoxia will be featured/discussed later in the paper but the figure callout should not be included. Could probably accomplish this through merging of that point into the prior sentence and eliminating the remainder.*

We will remove the callout to the figure in the revised version as suggested.

*Page 6, line 8. In this equation describing the biological source / sink terms of DO, it would be interesting to note what is done to ensure negative concentrations of dissolved oxygen are not achieved in the model.*

There is a specific treatment that allows DO not to reach negative values by switching off all processes consuming O2 below a $10^{-3}$ μmol.L$^{-1}$ threshold. This will be mentioned in the updated manuscript.

*Page 9, lines 1-2. The text here is a bit confusing. At the top of the paragraph it is stated: "… we used the standard Dipole Mode Index …". From the subsequent text in this section that indicates DMI based on model SST is calculated, I think what is meant is " … the standard DEFINITION of Dipole Mode Index …" is used.*

Corrected. We will provide correlations with the DMI derived both from the model and the observations.

*Page 9, line 6. " to the choice of either of them" is awkward. "which is chosen" would work better.*

This will be corrected, thanks.

*Page 9, line 26 -> top page 10. In the model during MAM, Fig. 6e shows that OCD is not really uniform in eastern AS; and OCD off Mumbai is comparable in value to OCD at bottom of the subcontinent. This directly contrasts what is described in the text. It also dampens the scenario described as a clear-cut influence of coastal KW propagation northward of the shoaled OCD / TCD condition that initiates in the STI region. I believe that the issue here is that care needs to be taken to make clear that description is centered on WCI to STI region and not relevant north of 10° where alongshore wind forcing is distinct from what exists farther south.*

This description will be improved as follows: "The tight relation between the OCD and TCD in the eastern Arabian Sea is further illustrated in Fig. 5, which displays the observed and modelled OCD and TCD seasonal climatologies. During the spring inter monsoon (March-May), the TCD and OCD are spatially quite uniform and deep (~100 m) in the southeastern Arabian Sea (Figs. 5a and 5e). This is also the case along the WCI south of 15°N despite the local upwelling favorable alongshore winds, indicating a remote control of OCD and TCD variations there. In contrast, the shallow OCD/TCD near the STI during this season is consistent with upwelling favorable winds in this region (Figs. 5a and 5e and dashed curve on Fig. 7a)."

*The referee comments are italicized.* Answers in regular typeface. Actions taken in red.

*Page 10, line 8. I think it is stretching what can be elucidated from the model results and presented figures to say that the OCD/TCD pattern is clearly suggestive of faster planetary wave propagation at lower latitudes. Lag associated with coastal KW propagating northward could also factor in.*

We agree and will modify the text accordingly.

*Page 10, bottom paragraph. The linkage of offshore OMZ to low coastal DO is again a thread in this part of text (see earlier remarks on my reservations).*

This discussion will be considerably strengthened in the updated manuscript by following the reviewer's suggestion. Figure 7 will provide a comparison of the seasonal oxygen and temperature evolution offshore and at the coast and of the remote and local alongshore winds. It shows that both oxygen and temperature seasonal variations along the coast from CaTS in-situ measurements match the corresponding offshore variability derived from the WOA and model data, suggesting a strong connection between the two. The seasonal wind evolution further allows discussing the respective influences of local and remote forcing (from the southern tip of India) in driving these variations.

*Page 11, lines 3 - 17, and figure 9. There are a number of interesting features in figure 9. Some comment on the inverse correlation regions appearing in 9a would be interesting to include and may provide some mechanistic insight to IOD-associated biophysical interaction that is the crux of this analysis. Some comment on the several instances where model TCD and observed SLA are out of phase in the 2002-2006 period (9b) would also be potentially illuminating (model issue?, sensitivity to WCI box definition?). Assessing any limitations in either of these is key to explore and characterize for the reader to fully trust results stemming from this analysis.*

This figure and the related text will be revised in the updated version to better discuss the physical mechanisms driving the interannual variations along the west coast of India. Figure 9 will be split in to two separate Figures. The new Figure 8 now includes previous Figure 9a along with timeseries of interannual TCD and OCD variations in the WCI box, to further highlight the strong physical control of interannual O2 variations in this region. The new Figure 9 now includes previous Figure 9b along with two additional maps. The new Figure 9b displays the correlation pattern of interannual model TCD in the northern Indian Ocean onto the timeseries of fall model TCD in the WCI box. New Figure 9c shows a similar analysis but for observed SLA. This Figure allows highlighting the good match between the interannual temperature variations along the WCI, and also allows to show that these variations are strongly related to temperature and wind variations at the southern tip of India (box in new Figure 9b,c), further demonstrating the role of remote wind fluctuations at the southern tip in driving the west coast interannual variations through coastal Kelvin wave propagation. The good match between timeseries of interannual STI wind fluctuations and both WCI modelled TCD and observed SLA (-0.65 and -0.69 correlation respectively) further demonstrates this connection. Despite the generally very accurate model behavior, we also added a brief discussion on the periods where the model agreement is weaker (2002-2006). Finally, the TCD and SLA patterns shown on Figure 9bc are also reminiscent of the typical IOD pattern. The relationship with IOD is further illustrated on Figures 10 and 11.

*Page 11, lines 19-20. Details of how anomalies are determined would be useful to document; caption of Fig. 10 is an option if this is not substantial enough to stand alone in methods section. Also, on line 20 panels a-c are noted as regression maps, which is contradictory to the information in the Fig. 10 caption (and plot labeling) that states panel c is a correlation map. For both of these data reductions details of how they are performed are not documented, making it problematic for the reader to correctly self-determine his/her interpretation. As for grasping why distinct analysis method was to zonal wind stress relative to how the other variables (OCD, TCD, SST) were treated, that is even more of a challenge for the reader to intuit.*

*The referee comments are italicized.* Answers in regular typeface. Actions taken in red.

Interannual anomalies for all variables have been calculated from monthly time-series by subtracting the mean seasonal cycle and applying a 3-month smoothing to remove the sub-seasonal variations. This will be explicitly stated in the Data and Method section. The panel with correlations is indeed confusing and anyway not necessary: we removed it and provided the regressed wind stress anomalies on panel a.

*Page 11, lines 19 - 23 and caption for Fig. 10. The terminology used in referring to the derived fields described here, and the terminology in the captions that accompany the associated distributions in Fig. 10, is inconsistent. Specifically, the narrative notes that all variables are internally varying anomalies but in the caption this is not obvious. In the caption, the data shown in panels a-c are noted to be regressed against "normalized oxycline interannual anomalies averaged over WCI box". My interpretation is that this describes the ts shown in Fig. 11a. If this is indeed the case, noting that as such would be very helpful to the reader as a way to better grasp what is presented in Fig. 10. This may entail modifying figure ordering w/in the ms.*

First, we agree that the terminology used in the caption of Figure 10 and in the related text were inconsistent. The bottom panels with correlations have been removed and regressed wind stress anomalies have been added on top panels. In addition, the WCI OCD time series used to obtain the regression maps shown on Figure 10 is now plotted in Figure 8b. This will be now clearly stated both in the modified text and in the caption of Figure 10.

*I find it interesting that the regression of OCD with OCD(WCI) in the WCI box (panel 10a) is actually relatively low compared to elsewhere in the IO domain. I think some interpretive commentary from the authors would be very interesting to see. Further, as I questioned earlier, does this have implications for the sensitivity / utility / robustness of the WCI box as a foundational component of this analysis? I would like to see the authors critically assess and comment on the choices they have made in setting up and carrying out their analysis.*

We will first add a couple of additional sentences to explain more clearly the meaning of these regression maps, i.e. that they provide the basin scale typical anomalies corresponding to an anomalously deep OCD in the WCI, which are indeed similar to those associated with a high DMI, i.e. a positive IOD. It is true that the regression coefficients of OCD in the WCI box are rather low (~6 m) as compared to the eastern equatorial Indian Ocean (~15m). This reflects the fact that the amplitudes of interannual OCD/TCD fluctuations are generally weaker in this region compared to other places. However, these small fluctuations have a tremendous impact on the ecosystem and related fisheries and hence deserve to be better understood. It must be noticed that, despite the modest WCI regression coefficient, corresponding correlation coefficients from this region to the southern tip of India are very large, demonstrating that the variability in the WCI box is representative of the OCD/TCD variations all along the western and southern coasts of India. The main conclusions of the paper do not change for a slightly different choice of the boundaries of the WCI box, as now noted when introducing this box.

*Page 11, line 24. For clarity and to benefit the reader, please explicitly associate / define how shallower / deeper OCD anomaly relates to + / - OCD anomaly.*

This will be clarified.

*Page 11, lines 25-31. The KW propagation patterns noted here are consistent with what is reported in the literature. However, the pathways and timings that are discussed are not identifiable in the Fig. 10. Appropriate referencing of the literature should be given to support what is stated. The remote forcing aspect of thermocline dynamics in the northern IO is quite complex; I strongly encourage the*

*authors to make the effort to clearly articulate and document what is known and how it relates to their study. I believe this would be highly appreciated by the IO readership.*

Appropriate referencing of the literature will be provided in this paragraph as follows: "Positive OCD anomalies along the WCI are also related to shallower OCD and TCD in the eastern Indian Ocean and along the eastern rim of the Bay of Bengal (i.e. negative anomalies). The associated large-scale wind patterns (Fig. 10c) explain these interannual OCD and TCD patterns. An anomalously deep OCD along the WCI is usually associated with anomalous easterlies at the equator that force an equatorial upwelling Kelvin wave and shoal the OCD and TCD in the eastern equatorial Indian Ocean. This equatorial Kelvin wave further propagates around the rim of the Bay as a coastal upwelling Kelvin wave, thereby shoaling the TCD and OCD there (e.g. McCreary et al., 1990; McCreary et al. 1996; Aparna et al., 2012; Suresh et al., 2016). Similar to what happens at the seasonal scale, zonal wind stress anomalies in the vicinity of Sri Lanka and the STI force a downwelling coastal Kelvin wave that propagates poleward along the western Indian coastline (e.g., Suresh et al., 2016), resulting in a deepening of the TCD and OCD there. The strong negative correlation between the WCI OCD and STI zonal winds interannual fluctuations (-0.73; Fig. 11b) further illustrates the strong influence of these winds in driving OCD and TCD interannual fluctuations in the southeastern part of the Arabian Sea."

*Page 11, line 31. Should specify that the correlation referred to here is negative.*

Done.

*Page 12, lines 4-8. The correlation distribution pattern for taux (Fig. 10c) is different enough that I would hesitate to even characterize it as "reminiscent of IOD signature"; in particular the sign shift in correlation that is apparent in the NE Bay of Bengal and eastward / off equator shift of strongest correlation (i.e., away from Sumatra coast where + IOD signature is most pronounced). Note as well that this is positive IOD signature. And related to that call for clarification, it would be nice to include an example of negative IOD manifestation to accompany the example of positive IOD (Fig. 5d).*

Earlier figure 10c has been removed (cf earlier comment). Several diagnostics in the paper point towards an influence of IOD on OCD/TCD variations in the WCI: (1) basin-scale wind stress patterns associated with anomalous OCD along the west coast of India (now on upper panels of updated Figure 10) look very similar to those associated with the DMI (i.e. a way to estimate the IOD wind signature), and we now state the pattern correlation of 0.85, (2) the new correlation patterns provided in Figure 9bc also resemble the IOD signature on thermocline and SLA variability and (3) both observed and model-derived DMI correlate well (0.62 and 0.67) with WCI interannual OCD variations. We thus think that we can state that this pattern is "reminiscent of the positive IOD signature". The asymmetries between positive and negative IODs are explicitly discussed later in the text based on Figure 13.

*Page 12, line 6. Here, the 10 d-f sequence is collectively referred to as regression maps, which is inconsistent with labeling / reporting elsewhere in narrative.*

The modifications in the figures of the revised version address this point.

*Page 12, lines 8-9. I think it is a stretch to make the statement that these figures, in particular panel 10d which is the key for illustrating the point, demonstrate a "strong link" between OCD in WCI region and IOD. In my view the level of regression in 10d for the WCI region is marginal and certainly much less pronounced than elsewhere (e.g., the Sri Lanka Dome).*

The "strong" was indeed a too "strong" word. We now explicitly state the 0.62/0.67 correlation between OCD in the WCI region and the observed/model DMI, and state that it is significant at the 99% level. More explicitly, we now explain that basin scale OCD, TCD and wind stress patterns

associated with OCD anomalies in the WCI region look very similar (and we now mention pattern correlations to support this) to those associated with the IOD (i.e. obtained through a regression with the DMI). This clearly supports the link we mention, although we agree it was not explained clearly enough in the previous version of the manuscript. The level of regression of Fig. 10d is an indication of the amplitude of the signal related with the IOD (which is of course stronger in the equatorial region) but does not provide direct information about how much the IOD control the variability there. Correlation provides this kind of information (the correlation squared being the level of OCD/TCD variance explained by the IOD).

*Page 12, lines 10-11. The correlation between model-based DMI and OCD(WCI) is interesting and suggestive of what the authors are trying to demonstrate (i.e., causal link between OCD and DMI). I would strongly suggest that this correlation also be performed with standard DMI product, for a couple of reasons. While the authors did report earlier in the ms that the standard DMI and their model-based DMI were largely similar and choice of which is applied did not substantially affect the results, performing the correlation with standard DMI provides grounding to a well-known established index and would mitigate any concerns that may/should arise for the reader vis a vis internal bias inherent in a model-model comparison.*

We now provide a comparison of the WCI OCD interannual variability with both model and observed DMI on this Figure. Because the observed and model DMI are strongly correlated (0.94), OCD interannual timeseries correlate well with both model DMI (0.67 correlation) and observed DMI (0.62). This will be elaborated in the updated manuscript. To further strengthen the physical explanation of how the IOD can influence the OCD variations along the WCI, Figure 11bc also provide the timeseries of the local and remote alongshore wind stress forcing: this analysis clearly shows that the interannual OCD variations along the WCI are far more correlated with STI winds (-0.73) than with local alongshore winds (-0.25).

*Page 12, lines 14-21, and Fig. 11b. This is an interesting distillation of results, though I still have reservations of robustness similar and related to what has been noted in comments above. I think it would be useful to find a way to split out the DMI values such that those that pass threshold and can be classified as positive / negative IOD are distinct from those that do not (i.e., normal and IOD states are clearly delineated). The regressions that are shown do not add much insight, but if they are retained then they should be performed only on points that are +/- IOD states and not inclusive of all positive of negative values.*

The updated Figure 12 (previous Fig. 11b) now highlights with different color codes the years corresponding to positive and negative IOD events (red and blue dots respectively) along with neutral years (black dots). Positive (resp. negative) IOD events are identified as years when the DMI index exceeds one standard deviation (resp. is lesser than minus one standard deviation). All other years are classified as normal years. Regression lines have been removed from the revised manuscript, as they did not provide much insight, as pointed out by the reviewer.

*However, even doing this, I am skeptical that particularly insightful result will be obtained. For the data that are shown that do represent + /- IOD occurrence, there is not really an associated systematic deepening / shoaling of OCD. Certainly the negative IOD case has a number of either OCD result. But there is also a positive IOD case with (slightly) shallowing OCD. It also appears that the years with observed and notable anoxic events tend to be during negative DMI (thought not necessarily negative IOD) but there is also a positive DMI case and a case that is almost uniformly normal (i.e., DMI _ 0 and OCD _ 0). Given these results, I do consider that the data are suggestive of positive IOD events influencing appearance of low oxygen in WCI but would be hesitant to make categorical conclusions. Overall, I think the authors need to be more even handed when reporting what is revealed in Fig.11b.*

New figure 13 (old Figure 12) clearly demonstrates that, on an average, positive IOD events lead to a deepening of the TCD/OCD along the west coast of India. Similarly, negative IOD events are

*The referee comments are italicized.* Answers in regular typeface. Actions taken in red.

associated, in general, with a shoaling of TCD/OCD along the WCI. However, although the correlation between DMI and WCI OCD is significant (0.67), it is not close to one, demonstrating that there are clear departures from this average picture. This is actually what is shown on new Figure 12 (previous Figure 11b), where positive IOD can be related sometimes with insignificant OCD anomalies and negative IOD can even be related in some instances with OCD deepening. This suggests that IOD is not the only process that controls the interannual OCD variations along the WCI. Other processes, such as local wind fluctuations or biological processes can indeed counteract the IOD influence. This discussion has been expanded in the revised manuscript. We now clearly state that "positive IODs suppress favourable conditions for anoxic events along the WCI and negative or neutral IODs are a necessary but not sufficient condition for anoxia to occur along the WCI ."

*Page 12, lines 21-27, and Fig. 12. This figure is also quite interesting. It suggests that both IOD phases lead to positive OCD and TCD displacement during SON, although for negative IOD this may not be statistically significant. What I find curious is that taux during SON in STI that has opposite sign between positive and negative IOD. Would this not lead to opposite oceanic response (i.e., upwelling vs. downwelling) between the two IOD phases, which then presumably has contrasting impact on thermocline displacement if this response translates northwards as coastal KW as authors have argued? The authors do note that taux condition for negative IOD is not as "robust" as that for positive IOD, however the relative values of anomaly for October (highest magnitude for SON period) are not strikingly distinct. Which begs the question as to how much OCD / TCD response one can ascribe to STI wind stress condition.*

First, we believe that there is a misunderstanding here. In this Figure, the negative IOD composite time series has been multiplied by minus one to ease comparison with positive events (as stated in the Figure caption). Hence, positive IOD events are associated with OCD and TCD deepening and negative zonal wind stress anomalies at the STI. In contrast, negative IOD events are associated with OCD/TCD shoaling (although hardly significant) and positive zonal wind stress anomalies at the STI. These evolutions are hence consistent with a Kelvin wave response at the STI to zonal wind variations associated with the IOD fluctuations. To avoid confusion, we now mention explicitly in the main text, figure caption and figure label (« -1 x Negative-IOD ») that the the negative IOD composite time series has been multiplied by minus one to ease comparison with positive events. To better asses how much OCD response can be ascribed to STI wind stresss variations, the new Figure 11b compares interannual time series of WCI OCD and STI Taux: these two parameters display a strong and significant correlation (-0.73), demonstrating that more than 50% of the OCD variability along the WCI can be attributed to STI wind variations. This will be more clearly stated in the updated text.

*Page 12, slide 30. Instead of "identified fifteen years ago", reiterating citation of relevant source material would be best.*

This will be done.

*Page 20, line 8. Title for this reference is incorrect.*

This will be corrected.

*Comments on Figures*
*General comment. There are several figures that include isolines superimposed on a second variable where a color scale is applied. In almost all cases, there is a need for additional labeling of the superimposed contour lines on the plot so these distributions can be grasped. There is also a need to note the contour intervals in the associated captions.*

This will be done.

*Figure 4. For panels 4b and 4d, the red contour line demarking thermocline is extremely difficult to*

*The referee comments are italicized.* Answers in regular typeface. Actions taken in red.

*see. In addition, the isotherms at depth (DO < 80 microM) are also difficult to see. For panels 4a and 4c, the 15_N line is difficult to spot as well.*

These colors have been changed to white.

*Figure 9. In panel 9a, white color in the IO domain has non-unique meaning. It can be either masked data or the transition from positive to negative correlation value. A way of uniquely distinguishing these should be used. Additionally, the white land mask is also not ideal but does have benefit of land-sea boundary.*

We now use grey shade for the mask.

*Panels 11a and 12 a-c. The combination of blue and black line colors is ineffective. Very difficult to distinguish between these two.*

Black has been changed to red for better visual appeal

**Revised figures**

[Figure]

**Revised Figure 6.** Seasonal evolution of oxygen (μmol.l$^{-1}$; color) and temperature ($^o$C; contour) vertical profiles averaged over the WCI box (indicated as black frame on Fig. 4) from (a) WOA13 and (b) model . The oxycline and thermocline depths are marked by thick red and black lines respectively in both panels.

*The referee comments are italicized.* Answers in regular typeface. Actions taken in red.

[Figure]

**Revised Figure 7. (a)** Monthly climatological timeseries of alongshore wind stress variations in the WCI box (thick black curve) and zonal wind stress at the southern tip of India (STI; dashed black curve) with positive values denoting the upwelling-favourable winds. Monthly climatological timeseries of upper ocean (0-40 m depth) averaged oxygen (blue) and temperature (red) **(b)** from the model (continuous) and WOA13 (dashed) in the WCI box and **(c)** from the in-situ CaTS data. Vertical bars on panel b and c indicate +/- one standard deviation around the mean value (displayed for CaTS data only when the number of years sampled for a given month exceeds five). Percentage of profiles for each calendar month for which **(d)** oxygen concentrations below 80 μmol.l⁻¹ occur within the top 50 m at WCI box in the model and **(e)** oxygen concentrations below 20 μmol.l⁻¹ occur within the top 20 m in CaTS data.

*The referee comments are italicized.* Answers in regular typeface. Actions taken in red.

[Figure]

**Revised Figure 8. (a)** Map of correlation between the modelled oxycline and thermocline depths interannual anomalies. **(b)** Fall interannual anomalies of the model TCD (m; red line) and OCD (m; black line) averaged over the WCI box (see frame on panel a). On panel a, values are grey shaded when the oxycline and/or thermocline could not be defined for more than 20% of the profiles at a given location

*The referee comments are italicized.* Answers in regular typeface. Actions taken in red.

[Figure]

**Figure 9. (a)** Fall interannual anomalies of the model thermocline depth (red line) and the altimetry-derived sea-level (blue line) averaged over the WCI box (see black frame on panels b and c) along with fall interannual zonal wind stress anomalies (dashed line) averaged over the STI box (see dashed frame on panels b and c). **(b)** Correlation pattern of fall interannual thermocline anomalies on to that averaged over the WCI box in the model. **(c)** Same as (b) but for altimetry-derived sea-level anomalies.

[Figure]

**Revised Figure 10**. Regression patterns of fall interannual anomalies of modelled **(a)** oxycline (m; shaded) and wind stress (N.m⁻²; vectors), **(b)** thermocline (m; shaded) and SST (°C; contours with 0.1°C interval) onto the fall oxycline interannual anomalies averaged over WCI box normalized by its standard deviation. **(c-d)** Same as **(a-b)** but regressed onto the observed fall DMI index.

[Figure]

**Revised Figure 11. (a)** Fall time series of interannual anomalies of modelled oxycline depth averaged over the WCI box (black line) and modelled (continuous red) and observed (dashed red) fall DMI, **(b)** Fall time series of interannual anomalies of modelled oxycline depth averaged over the WCI box (black line) and zonal wind stress averaged over the southern tip of India (red line; STI box shown as dashed frame on Figure 9bc). **(c)** Fall time series of interannual anomalies of modelled oxycline depth (black line) and alongshore wind stress (red line) averaged over the WCI box. Upwelling-favourable winds on panel b and c are positive. Correlation coefficients between the variables are given on each panel.

[Figure]

**Revised Figure 12.** Scatterplot of the fall interannual anomalies of WCI oxycline depth against DMI. Years of anoxic events reported by Naqvi et al. (2009) are marked as stars along with the corresponding years. Red and blue symbols respectively indicate positive and negative IOD events (defined as events where DMI exceed one standard deviation).

[Figure]

**Figure 13.** Seasonal evolution of anomalous composites of **(a)** WCI OCD, **(b)** WCI TCD and **(c)** STI alongshore wind stress during positive (red) and negative (blue) IOD events. Positive (negative) IODs are defined when the DMI averaged for fall season is greater (less) than 1° C. The whiskers indicate the 95% confidence interval on the composited value. Positive IOD events considered in this composite are years 1961, 1963, 1967, 1972, 1977, 1982, 1994, 1997, and 2006 while negative ones are 1964, 1973, 1974, 1975, 1979, 1981, 1984, 1992, 1996, 1998, and 2010.

---

## Author Comment (AC2) · 28 Oct 2016

Dear Editor,

Please find our replies to the review comments in the attached file.

We thank the referees for their excellent review and critical comments.

Thank you, Sincerely, Suresh

Please also note the supplement to this comment:
http://www.biogeosciences-discuss.net/bg-2016-195/bg-2016-195-AC2-supplement.pdf

---

## Author Response (AR1)

*The referee comments are italicized.* Answers in regular typeface. Actions taken in red.

**Answers to referee 1 comments**

*Anonymous Referee #1:*
*Summary: The paper by Parvathi et al. investigates the seasonal and interannual variability of dissolved oxygen off the West Coast of India (WCI) using a coupled physical-biogeochemical regional simulation of the North Indian Ocean. More specifically, the study documents the recent variability in the oxycline along the WCI and explores its potential drivers. It is shown that the seasonal as well as the year-to-year fluctuations in the depth of hypoxia (and hence the likelihood of occurrence of anoxic events in the near-surface coastal waters) are controlled to a large extent by the variability of the depth of the thermocline, and hence are essentially physically driven. Authors show that oxycline like thermocline is shallowest in fall, thus eventually allowing anoxia to develop and reach the coastal region during this period of the year. Finally, it is shown that the year-to-year fluctuations in the oxycline (and hence the potential severity of hypoxia) in the WCI region is partially controlled by the Indian Ocean Dipole (IOD). This is because of the sensitivity of the thermocline in this region to coastal Kelvin waves generated by easterly wind anomalies associated with IOD. Yet this effect is only strong during positive IODs where the downwelling Kelvin waves generated by easterlies near the southern tip of India lead to a deepening of the oxycline that prevents anoxia. In contrast, during negative IODs the effect is of smaller amplitude and hence is less important. The authors conclude that these findings have important implications in terms of the predictability of fall anoxia along WCI.*

*General comment:*
*The subject of the paper is highly relevant in the general context of quantifying and better understanding the drivers of naturally occurring coastal anoxia and what controls its year-to-year variability. The paper is well written and presents important new findings. The experimental design is appropriate despite some limitations (e.g., model resolution). For these reasons I recommend the publication of this manuscript after improving the presentation and the discussion of certain aspects of the study, namely the potential role of biology in contributing to O2 variability and the implications of the model low-resolution for the conclusions of the study.*

We thank the referee for his encouraging review. As detailed below, we propose to improve the presentation and discussion of the potential role of biology in contributing to O2 variability and the implication of the low-resolution of the model used.

*Here below I detail my comments:*
*Specific comments:*
*Page 2, line 16: "The frequent anoxic conditions that occurred…"*

This has been corrected.

*Page 4, line 13: The 1/4 horizontal resolution does not allow a proper representation of coastal upwelling either. The underestimation or potential misrepresentation of coastal (wind-driven) upwelling in the model needs to be mentioned and its potential implications discussed.*

We briefly mention this model limitation in the introduction section but defer to the discussion section for more extensive discussion regarding the implications of the underestimation of coastal upwelling. The discussion on the model limitation has been considerably expanded and can now be found in the last section of the manuscript (see from P17 L18 to P18 L14).

*Page 5, lines 9-20: in addition to the limitations discussed by authors, the CaTS dataset comes from a single coastal site and hence may not be representative of the dynamics of the whole WCI. This needs to be stressed.*

*The referee comments are italicized.* Answers in regular typeface. Actions taken in red.

Thank you for pointing this out. This is now mentioned in the revised draft (P5, L22-23).

*Page 6, 2nd equation: all parameters ($R^1_{o:c}$, $R^2_{o:c}$, etc…) should be explicitly defined in the text.*

All parameters are now explicitly defined in the revised draft (P6 L22-26).

*Page 6, Is sediment respiration represented in the model? Please specify. Also, how are denitrification and anamox represented in the model?*

We agree with the reviewer that such a description was missing in the earlier version of the paper. The model indeed includes a very simple description of the sediment processes. The metamodel of Middelburg et al. (1996) is also used to compute the relative contribution of denitrification to the remineralization of the organic matter. Denitrification starts when oxygen falls below a threshold set to 6 µmol.L$^{-1}$: below this threshold, nitrate instead of oxygen starts to be increasingly consumed during the remineralization of organic matter. However, anamox is not represented in the model. To address the reviewer comment, a more thorough discussion of these modelling aspects have been included in the model description section of the paper (P6 L27 to P7 L4).

Related reference:

Middelburg, J. J., Soetaert, K., Herman, P. M., & Heip, C. H.: Denitrification in marine sediments: A model study. *Global Biogeochemical Cycles*, *10*(4), 661-673, 1996.

*Page 9, lines 14-15: I don't think the effect of biology on oxygen is small at the thermocline depth. The respiration fluxes can be very large at 100m. The fact that the oxycline and the thermocline show high correlations may result from the fact that biology itself is constrained by vertical physics and as a consequence that the nutricline and the thermocline are tightly coupled. This statement needs to be reformulated and the role of biology further discussed (authors may consider quantifying the individual contributions of biology and physical transport to oxygen variability).*

Resplandy et al. (2012) did already quantify the respective contributions of biology and vertical transport to the seasonal oxygen variability along the west coast of India. Despite the compensation of biological sink and dynamical source of oxygen on an annual average, they found that the seasonality in the dynamical transport of oxygen is 3 to 5 times larger than that in the biological sink. The seasonality in oxygen along the WCI hence primarily arises from the vertical displacement of the oxycline attributed to the influence of northward propagating coastal Kelvin waves forced remotely from the Bay of Bengal. This conclusion does not support the hypothesis of Sarma (2002), who suggested that the weak oxygen seasonality could arise from compensation between the dynamical input and the biological uptake of oxygen during each season. This hypothesis could not however be verified owing to the absence of reliable seasonal estimates of oxygen consumption (Sarma, 2002). The text has been modified to explicitly mention the dominance of vertical transport on the modelled seasonal oxygen variations in this region derived from the oxygen budget analysis of Resplandy et al. (2012) (see P10 L4-13).

Related references:

Resplandy, L., Lévy, M., Bopp, L., Echevin, V., Pous, S., Sarma, V. V. S. S., and Kumar, D.: Controlling factors of the oxygen balance in the Arabian Sea's OMZ, Biogeosciences, 9, 5095–5109, doi:10.5194/bg-9-5095-2012, 2012.
Sarma, V. V. S. S.: An evaluation of physical and biogeochemical processes regulating perennial suboxic conditions in the water column of the Arabian Sea, Global Biogeochem. Cy., 16, 1082, doi:10.1029/2001GB001461, 2002.

*Page 9, lines 18-22: It is not clear why the correlation between the OCD and the TCD is low in the southern and the southwestern parts of the domain. In the southwestern part authors mention a*

*possible role of lateral advection there, but it is not clear how this might affect the correlation between these two quantities.*

If any other process than vertical advection strongly contributes to OCD variability (i.e. lateral advection, biology, etc…), there is no reason for OCD variability to be in phase with TCD variability and correlation between the two variables should hence decrease.

*In the equatorial region authors suggest the low correlation may be due to the definitions used for oxycline and thermocline that may not be meaningful there. Yet, in section 2.4 (page 8, lines 28-31) it is stated that the results and the conclusions of the study are insensitive to the oxygen (50-150 mmol/m3) and temperature (20-25C) criteria used to define these two depths. These two statements appear to contradict each other.*

We agree that these two statements are misleading. We meant that main results regarding the variability of the OCD/TCD variations in the Arabian Sea and more specifically in our focus region along the WCI are not sensitive to that choice. This has been clarified in the revised manuscript that this holds for the northern Indian Ocean only (P9 L13-16).

*Page 9, lines 25-27: during the spring inter-monsoon, the (northwesterly) winds along the WCI are also upwelling-favorable. Yet, the thermocline is relatively deep there (especially in observations). This is an indication that TCD alone cannot be used as a proxy for wind-driven coastal upwelling (probably because of the non-local effects of the coastal Kelvin waves).*

We agree with the reviewer. This is now mentioned in the revised manuscript (P10 L23-26).

*Page 10, lines 4-5: in the northern part of the WCI, alongshore (southward) winds seem to be upwelling-favourable. Maybe plotting seasonal time series of upwelling index together with OCD and TCD would help figuring out how these three quantities co-vary (can be added to Fig 7).*

Following the reviewer concern, the paper has been revised to include a more thorough discussion of the local and remote wind control on the OCD and TCD variations along the west coast of India. The updated Figure 7 now provides a direct comparison between the wind seasonal variations along the coast and at the southern tip of India and the west coast of India oxygen seasonal variability in both the model and observations. This figure shows that the TCD starts to shallow in April and becomes shallowest in September-October, the peak season of the upwelling. Local alongshore winds along the WCI are favourable to upwelling only during the southwest monsoon, indicating the influence of remote wind forcing on the upwelling along the WCI. The text has been accordingly revised to describe this feature in more detail (P11 L10-30).

*Page 10, lines 24-32: the overestimation of oxygen in the WCI may not only be due to the coarse representation of the shelf dynamics but also the general tendency of the model to underestimate the intensity of the OMZ in the northern Arabian Sea (see Fig 4 for example). This needs to be discussed.*

We don't necessarily agree with that statement. The good agreement between the model and WOA seasonal oxygen variations in the WCI box (Fig. 7) suggests that the model is able to accurately capture the oxygen variations offshore the WCI. This good agreement suggests that the model OMZ underestimation in the northwestern AS (Fig. 3ac) does not affect the oxygen representation along the WCI. It is hence more likely that shelf-specific processes may explain the lower oxygen contents observed at CaTS as compared to the model and WOA data. These issues are in any case now addressed in the discussion section (P18 L1-14).

*Page 13, lines 7-8: Identifying the drivers of coastal anoxia requires further investigation. In particular, the role of biology and the effects of O2 consumption on the shelf need to be quantified and contrasted with the impact of vertical and lateral advection. It is not clear how much of anoxia is driven by large-scale advection of O2-depleted water (from the Arabian Sea OMZ) vs. local*

*The referee comments are italicized.* Answers in regular typeface. Actions taken in red.

*consumption of O2 due to respiration in the water column and on the shelf.*

We agree with the reviewer that this statement is too strong, given our model limitations. We tuned down that statement in the revised manuscript (P15 L7-8). In addition, a revised discussion on the need of a refined assessment of the large-scale and local biological processes influencing anoxia along the shelf is now provided in the discussion section (from P17 L29 to P18 L14).

*Page 13, lines 17-20: Any explanation for the asymmetry of the impacts of positive and negative IODs (and in particular the potential reason for why the wind anomalies are weaker during negative IODs)?*

Negative IODs tend to be weaker than their positive counterpart due to nonlinear response of the deep atmospheric convection to SST anomalies (Saji and Yamagata 2003; Hong et al. 2008; Cai et al. 2013). The differences in wind patterns (the fact that for a given DMI absolute value, positive IODs tend to display wind anomalies that extend more northward) may also be related to this asymmetry. This is now mentioned in the revised manuscript (P16 L9-16).

Related references:

Cai, W., Zheng, X. T., Weller, E., Collins, M., Cowan, T., Lengaigne, M., ... & Yamagata, T. (2013). Projected response of the Indian Ocean Dipole to greenhouse warming. *Nature Geoscience*, *6*(12), 999-1007.
Hong, C. C., Li, T., & Kug, J. S. (2008). Asymmetry of the Indian Ocean Dipole. Part I: Observational Analysis*. *Journal of climate*, *21*(18), 4834-4848.
Saji, N. H., & Yamagata, T. (2003). Possible impacts of Indian Ocean dipole mode events on global climate. *Climate Research*, *25*(2), 151-169.

*Page 15, lines 11-14: As acknowledged by the authors, the (relatively) coarse resolution of the model (1/4) is probably the main limitation of the study. The underestimated or misrepresented coastal upwelling and mesoscale eddies in the model may have important consequences. I encourage the authors to further discuss the implications this might have and how it may (or may not) affect their conclusions/results*

This has been done in the revised discussion (cf answers to your first comment).

*Page 15, lines 11-16: if the model does not represent sediment processes, the lack of benthic respiration could also limit the model ability to represent the dynamics of coastal hypoxia along the shelf. This needs to be clarified.*

The model does include a simple representation of sediment processes. The simplicity of this representation may however be a limit of the model's ability to represent the dynamics of coastal hypoxia. This will be clearly stated in the updated discussion of the potential role of biology that is quoted in answer to your comment above (from P17 L29 to P18 L14).

*Figure 3: the colorbar is missing.*

Sorry for this. Thank you for pointing this out. The colorbar has been added.

*Figure 3: since the study focus is on coastal anoxia that develops during fall, maybe it would be good to show how the model reproduces chlorophyll during this season.*

Figure R1 below provides the summer and fall Chl patterns for the model and observations. As shown in this Figure, the patterns during these two seasons share a lot in common, with a tendency for weaker Chl values in fall compared to summer in both model and observations. We hence did not

include the fall Chl pattern in the revised manuscript but will clearly state that the fall Chl patterns are similar, although weaker, than those in summer in both observations and model (P8 L25-26).

[Figure]

**Figure R1:** Northern Indian Ocean surface chlorophyll (mg.m$^{-3}$) climatology for (left) summer (June-August) and (right) fall (September-October) in (top) the ESA satellite product and (bottom) model.

**Answers to referee 2 comments**

*General comments: Ultimately, I think that there could be a linkage whereby positive IOD occurrence influences OCD conditions along the SW coast of India. The authors have taken a great stab at establishing this link but have fallen a bit short, in my opinion, of establishing the case. Significantly tightening up the analysis, following the suggestions I have made in specific comments below, would go a long way toward achieving a reasoned, well considered analysis that can at least strongly suggest such a linkage is present. It may be that additional data, refinement of this analysis in terms of key locations for exploring causality and higher resolution modeling with nested shelf regions, will be required to fully reveal what is tantalizingly indicated in the current effort.*

We thank the reviewer for his encouraging comments. We followed all reviewer's suggestions, which led to significant changes in the Figures list as summarized below:

1- As suggested by the reviewer, results from Figure 1 have been integrated in the new Figures 6 and 7. The text related to the earlier figure has hence been dropped from the introduction and a more precise discussion on the past literature discussing the offshore/shelf oxygen connection is now included in the introduction.

2- New Figure 4 provides a comparison between O2 and temperature seasonal variations in the WCI box for both model and WOA data.

3- The updated Figure 7 gathers results from old Figure 7 augmented with an assessment of local and remote (from the southern tip of India) seasonal wind variations. This updated Figure allows us discussing in more details: (1) the modelled and observed seasonal phasing between upwelling and oxygen depletion, pointing towards a strong physical control of seasonal oxygen variations, (2) the seasonal phasing between offshore and coastal oxygen variations, further highlighting the offshore/coastal connection at seasonal timescales and (3) the remote influence of STI wind seasonal variations on WCI upwelling variability.

4- The updated Figure 8 now includes timeseries of interannual oxycline and thermocline variations in the WCI box, further highlighting their in-phase relationship and hence the strong physical control onto WCI interannual oxycline variability.

5- The updated Figure 9 allows demonstrating the strong control of STI wind variability on observed and modelled WCI interannual thermocline variations through coastal Kelvin waves, along with the ability of the model to capture observed interannual thermocline variations in the WCI region.

6- The updated Figure 10 is a simplified version of previous Figure 10, from where the correlation between winds and the WCI oxycline / DMI index has been removed.

7- The updated Figure 11 demonstrates that (1) WCI interannual OCD variations are well correlated to the DMI index computed from either the model and observations and (2) that these variations are remotely controlled by interannual wind variations in the STI region rather than by local winds.

These new figures are appended below for your reference and the text will be changed accordingly. Below, we further provide point-by-point answer to each of your specific concerns.

*Specific comments :*

*The referee comments are italicized.* Answers in regular typeface. Actions taken in red.

*Introduction, p. 2 & 3. The bottom paragraph of p. 2 and top paragraph of p. 3 have diverted from providing background information of the research to be undertaken to an initial summarizing, drawn from figure 1, of aspects of the study that is being reported on. This leads to an awkward shifting between presenting problem framework and results that interferes with coherent, logically developed reporting. I would recommend that all call outs to Fig. 1 be eliminated from the Introduction section. This could be simply accomplished by shifting the two noted paragraphs out, but note that there is at least one additional callout to Fig. 1 to address. As to the content of these two paragraphs, I do have additional specific comments. In the first of these paragraphs, the authors first introduce the notion that the open ocean OMZ of the Arabian Sea has connectivity to the hypoxia that manifests along the west coast of India. From my knowledge of the literature, and from the literature the authors' have cited, this has so far only been posited and in my view is more anecdotal than proven fact. The 1/4_ model solution that the authors use as principal basis of their findings is challenged to demonstrate causality on this point. The authors should acknowledge this explicitly, make a more compelling case based on existing literature, or leverage this in from some of their higher resolution modelling efforts.*

The seminal paper by Banse (1959) shows a section across the shelf off Cochin located ~10$^o$N on the west coast of India. Based on the analysis of this section, Banse (1959) stated that the hypoxic conditions off Cochin were clearly connected with seasonal upwelling variations in relation with the southwest monsoon. Similar findings were reported off Bombay by Carruthers et al. (1959), who showed the layer of minimum oxygen was upsloping towards the coast right after the summer monsoon demise. Similar cross-shore sections along the west coast of India across the monsoon season are shown in Naqvi et al. (2009) and used to support the notion that it is the seasonal upwelling that brings to the west Indian continental shelf the oxygen-poor subsurface waters that then turn anoxic by late summer due to exhaustion of oxygen and nitrate by heterotrophic micro-organisms. Another recent paper (Gupta et al. 2016) further concluded that upwelling of oxygen-deficient waters during the monsoon is the major process regulating the biogeochemistry along this shelf using ten shelf transects during 2012 near 10°N. We believe that these observational analyses at different locations along the west coast of India are clear indications that there is some connection between open ocean seasonal oxygen variability and coastal hypoxia. We however agree with the reviewer that the discussion based on Figure 1 should rather belong to the result section. We also agree with the reviewer that the detailed processes by which these exchanges occur are not yet properly understood, nor resolved by our model or existing datasets. To address the reviewer's concerns, we have performed the following changes:

1- All callouts to Figure 1 has eliminated from the introduction. Results from this Figure has been incorporated in Figures 6 and 7 to discuss the connection between the offshore and costal seasonal O2 variability and the role of the remote and local seasonal alongshore wind variations in driving this variability in more detail.

2- We have rewritten the bottom paragraph of p2, to more precisely acknowledge on which observational basis the influence of upwelling has been reported as the major process driving the coastal hypoxia along the West Indian continental shelf. We especially refer to the results of Banse et al. (1959), Carruthers et al. (1959), Naqvi et al. (2009) and Gupta et al. (2016) who all support this offshore-coastal connection based on the analysis of the cross-shore transects data at different locations along the west coast of India. We however also clearly acknowledge in this paragraph that these conclusions were drawn from punctual cruises snapshots and that the processes through which these exchanges take place are unclear. Updated Figure 7 now provides a comparison of the seasonal oxygen evolution offshore and at the coast and of the remote and local alongshore winds. It allows clearly illustrating the strong seasonal phasing between the offshore and coastal evolution, further pointing towards a strong connection between open ocean OMZ and coastal hypoxia through the upwelling processes. This Figure also allows to demonstrate the remote control of the alongshore winds at the southern tip of India on the seasonal upwelling conditions along the west coast of India.

3- We have also clarify when introducing the model that it does not resolve the details of the

*The referee comments are italicized.* Answers in regular typeface. Actions taken in red.

shelf / open ocean interactions, but is merely used to understand the processes that control offshore variations (P4 L14-18). We have also included a discussion of the need for improved observations and modeling to resolve processes of shelf-open ocean exchanges in the discussion (P17 L17 to P18 L22).

*Page 5, line 20. In the last sentence, it is interesting to note that these observed instances of full anoxia will be featured/discussed later in the paper but the figure callout should not be included. Could probably accomplish this through merging of that point into the prior sentence and eliminating the remainder.*

We removed the callout to the figure in the revised version as suggested.

*Page 6, line 8. In this equation describing the biological source / sink terms of DO, it would be interesting to note what is done to ensure negative concentrations of dissolved oxygen are not achieved in the model.*

There is a specific treatment that allows model not to reach negative values by switching off all processes consuming $O_2$ below a $10^{-3}$ $\mu mol.L^{-1}$ threshold. This is now mentioned in the updated manuscript (P6 L27-28).

*Page 9, lines 1-2. The text here is a bit confusing. At the top of the paragraph it is stated: "… we used the standard Dipole Mode Index …". From the subsequent text in this section that indicates DMI based on model SST is calculated, I think what is meant is " … the standard DEFINITION of Dipole Mode Index …" is used.*

Corrected. We have now provided correlations with the DMI derived both from the model and the observations.

*Page 9, line 6. " to the choice of either of them" is awkward. "which is chosen" would work better.*

This sentence has been modified, thanks.

*Page 9, line 26 -> top page 10. In the model during MAM, Fig. 6e shows that OCD is not really uniform in eastern AS; and OCD off Mumbai is comparable in value to OCD at bottom of the subcontinent. This directly contrasts what is described in the text. It also dampens the scenario described as a clear-cut influence of coastal KW propagation northward of the shoaled OCD / TCD condition that initiates in the STI region. I believe that the issue here is that care needs to be taken to make clear that description is centered on WCI to STI region and not relevant north of 10° where alongshore wind forcing is distinct from what exists farther south.*

To address the reviewer comment, this description has been improved (P10 L22-27).

*Page 10, line 8. I think it is stretching what can be elucidated from the model results and presented figures to say that the OCD/TCD pattern is clearly suggestive of faster planetary wave propagation at lower latitudes. Lag associated with coastal KW propagating northward could also factor in.*

We agree. This is now mentioned in the updated version (P11 L1-2).

*The referee comments are italicized.* Answers in regular typeface. Actions taken in red.

*Page 10, bottom paragraph. The linkage of offshore OMZ to low coastal DO is again a thread in this part of text (see earlier remarks on my reservations).*

This discussion has been considerably strengthened in the updated manuscript by following the reviewer's suggestion (see P11 L10-29). Figure 7 now provides a comparison of the seasonal oxygen evolution offshore and at the coast and of the remote and local alongshore winds. It shows that oxygen seasonal variations along the coast from CaTS in-situ measurements match the corresponding offshore variability derived from the WOA and model data, suggesting a strong connection between the two. The seasonal wind evolution further allows discussing the respective influences of local and remote forcing (from the southern tip of India) in driving these variations.

*Page 11, lines 3 - 17, and figure 9. There are a number of interesting features in figure 9. Some comment on the inverse correlation regions appearing in 9a would be interesting to include and may provide some mechanistic insight to IOD-associated biophysical interaction that is the crux of this analysis. Some comment on the several instances where model TCD and observed SLA are out of phase in the 2002-2006 period (9b) would also be potentially illuminating (model issue?, sensitivity to WCI box definition?). Assessing any limitations in either of these is key to explore and characterize for the reader to fully trust results stemming from this analysis.*

This figure and the related text has been revised in the updated version to better discuss the physical mechanisms driving the interannual variations along the west coast of India (from P12 L15 to P13 L10). Figure 9 is split into two separate Figures (new Figure 8 and 9). The new Figure 8 now includes previous Figure 9a along with timeseries of interannual TCD and OCD variations in the WCI box, to further highlight the strong physical control of interannual oxygen variations in this region. The new Figure 9 now includes previous Figure 9b augmented with time series wind variations at the southern tip of India and two additional maps. The new Figure 9b displays the correlation pattern of interannual model TCD in the northern Indian Ocean onto the timeseries of fall model TCD in the WCI box. New Figure 9c shows a similar analysis but for observed SLA. This Figure allows highlighting the good match between the interannual thermocline variations along the WCI. The good match between timeseries of interannual wind fluctuations at the southern tip of India provided on Figure 9a and both WCI modelled TCD and observed SLA (-0.65 and -0.69 correlation respectively) further demonstrates the role of remote wind fluctuations at the southern tip in driving the west coast interannual variations through coastal Kelvin wave propagation. Despite the generally very accurate model behavior, we also mention the periods where the model agreement is weaker (2002-2006). Finally, the TCD and SLA patterns shown on Figure 9bc are also reminiscent of the typical IOD pattern. The relationship with IOD is further illustrated on Figures 10 and 11.

*Page 11, lines 19-20. Details of how anomalies are determined would be useful to document; caption of Fig. 10 is an option if this is not substantial enough to stand alone in methods section. Also, on line 20 panels a-c are noted as regression maps, which is contradictory to the information in the Fig. 10 caption (and plot labeling) that states panel c is a correlation map. For both of these data reductions details of how they are performed are not documented, making it problematic for the reader to correctly self-determine his/her interpretation. As for grasping why distinct analysis method was to zonal wind stress relative to how the other variables (OCD, TCD, SST) were treated, that is even more of a challenge for the reader to intuit.*

Interannual anomalies for all variables have been calculated from monthly time-series by subtracting the mean seasonal cycle and applying a 3-month smoothing to remove the sub-seasonal variations. This has been explicitly stated in the Data and Method section (P9 L19-20). The panel with correlations is indeed confusing and anyway not necessary: we removed it and provided the regressed wind stress anomalies on panel a.

*The referee comments are italicized.* Answers in regular typeface. Actions taken in red.

*Page 11, lines 19 - 23 and caption for Fig. 10. The terminology used in referring to the derived fields described here, and the terminology in the captions that accompany the associated distributions in Fig. 10, is inconsistent. Specifically, the narrative notes that all variables are internally varying anomalies but in the caption this is not obvious. In the caption, the data shown in panels a-c are noted to be regressed against "normalized oxycline interannual anomalies averaged over WCI box". My interpretation is that this describes the ts shown in Fig. 11a. If this is indeed the case, noting that as such would be very helpful to the reader as a way to better grasp what is presented in Fig. 10. This may entail modifying figure ordering w/in the ms.*

First, we agree that the terminology used in the caption of Figure 10 and in the related text were inconsistent. The bottom panels with correlations have been removed and regressed wind stress anomalies have been added on top panels. In addition, the WCI OCD time series used to obtain the regression maps shown on Figure 10 is now plotted in Figure 8b. This is now clearly stated both in the modified text and in the caption of Figure 10.

*I find it interesting that the regression of OCD with OCD(WCI) in the WCI box (panel 10a) is actually relatively low compared to elsewhere in the IO domain. I think some interpretive commentary from the authors would be very interesting to see. Further, as I questioned earlier, does this have implications for the sensitivity / utility / robustness of the WCI box as a foundational component of this analysis? I would like to see the authors critically assess and comment on the choices they have made in setting up and carrying out their analysis.*

We first added a couple of additional sentences to explain more clearly the meaning of these regression maps, i.e. that they provide the basin scale typical anomalies corresponding to an anomalously deep OCD in the WCI, which are indeed similar to those associated with a high DMI, i.e. a positive IOD (P13 L12-14). It is true that the regression coefficients of OCD in the WCI box are rather low (~6 m) as compared to the eastern equatorial Indian Ocean (~15m). This reflects the fact that the amplitudes of interannual OCD/TCD fluctuations are generally weaker in this region compared to other places. However, these small fluctuations have a tremendous impact on the ecosystem and related fisheries and hence deserve to be better understood. It must be noticed that, despite the modest WCI regression coefficient, corresponding correlation coefficients from this region to the southern tip of India are very large (see Figure 9bc), demonstrating that the variability in the WCI box is representative of the OCD/TCD variations all along the western and southern coasts of India. The main conclusions of the paper do not change for a slightly different choice of the boundaries of the WCI box, as now noted when introducing this box (P9 L18-19).

*Page 11, line 24. For clarity and to benefit the reader, please explicitly associate / define how shallower / deeper OCD anomaly relates to + / - OCD anomaly.*

This will be clarified (P13 L16, L18).

*Page 11, lines 25-31. The KW propagation patterns noted here are consistent with what is reported in the literature. However, the pathways and timings that are discussed are not identifiable in the Fig. 10. Appropriate referencing of the literature should be given to support what is stated. The remote forcing aspect of thermocline dynamics in the northern IO is quite complex; I strongly encourage the authors to make the effort to clearly articulate and document what is known and how it relates to their study. I believe this would be highly appreciated by the IO readership.*

Appropriate referencing of the literature are now provided in this paragraph (P13 L12-30).

*Page 11, line 31. Should specify that the correlation referred to here is negative.*

*The referee comments are italicized.* Answers in regular typeface. Actions taken in red.

Done.

*Page 12, lines 4-8. The correlation distribution pattern for taux (Fig. 10c) is different enough that I would hesitate to even characterize it as "reminiscent of IOD signature"; in particular the sign shift in correlation that is apparent in the NE Bay of Bengal and eastward / off equator shift of strongest correlation (i.e., away from Sumatra coast where + IOD signature is most pronounced). Note as well that this is positive IOD signature. And related to that call for clarification, it would be nice to include an example of negative IOD manifestation to accompany the example of positive IOD (Fig. 5d).*

Earlier figure 10c has been removed (cf earlier comment). Several diagnostics in the paper point towards an influence of IOD on OCD/TCD variations in the WCI: (1) basin-scale wind stress patterns associated with anomalous OCD along the west coast of India (now on upper panels of updated Figure 10) look very similar to those associated with the DMI (i.e. a way to estimate the IOD wind signature), and we now state that the the OCD pattern correlation is 0.85, (2) the new correlation patterns provided in Figure 9bc also resemble the IOD signature on thermocline and SLA variability and (3) both observed and model-derived DMI correlate well (0.62 and 0.67) with WCI interannual OCD variations. We thus think that we can state that this pattern is "reminiscent of the positive IOD signature". The asymmetries between positive and negative IODs are explicitly discussed later in the text based on Figure 13.

*Page 12, line 6. Here, the 10 d-f sequence is collectively referred to as regression maps, which is inconsistent with labeling / reporting elsewhere in narrative.*

The modifications in the figures of the revised version address this point.

*Page 12, lines 8-9. I think it is a stretch to make the statement that these figures, in particular panel 10d which is the key for illustrating the point, demonstrate a "strong link" between OCD in WCI region and IOD. In my view the level of regression in 10d for the WCI region is marginal and certainly much less pronounced than elsewhere (e.g., the Sri Lanka Dome).*

The "strong" was indeed a too "strong" word. We now explicitly state the 0.62/0.67 correlation between OCD in the WCI region and the observed/model DMI, and state that it is significant at the 99% level. More explicitly, we now explain that basin scale OCD, TCD and wind stress patterns associated with OCD anomalies in the WCI region look very similar (and we now mention pattern correlations to support this) to those associated with the IOD (i.e. obtained through a regression with the DMI). This clearly supports the link we mention, although we agree it was not explained clearly enough in the previous version of the manuscript. The level of regression of Fig. 10d is an indication of the amplitude of the signal related with the IOD (which is of course stronger in the equatorial region) but does not provide direct information about how much the IOD control the variability there. Correlation provides this kind of information (the correlation squared being the level of OCD/TCD variance explained by the IOD).

*Page 12, lines 10-11. The correlation between model-based DMI and OCD(WCI) is interesting and suggestive of what the authors are trying to demonstrate (i.e., causal link between OCD and DMI). I would strongly suggest that this correlation also be performed with standard DMI product, for a couple of reasons. While the authors did report earlier in the ms that the standard DMI and their model-based DMI were largely similar and choice of which is applied did not substantially affect the results, performing the correlation with standard DMI provides grounding to a well-known established index and would mitigate any concerns that may/should arise for the reader vis a vis internal bias inherent in a model-model comparison.*

We now provide a comparison of the WCI OCD interannual variability with both model and observed DMI on this Figure. Because the observed and model DMI are strongly correlated (0.94), OCD interannual timeseries correlate well with both model DMI (0.67 correlation) and observed DMI (0.62). This is now discussed in the updated manuscript (P14 L5-9). To further strengthen the physical

*The referee comments are italicized.* Answers in regular typeface. Actions taken in red.

explanation of how the IOD can influence the OCD variations along the WCI, Figure 11bc also provide the timeseries of the local and remote alongshore wind stress forcing: this analysis clearly shows that the interannual OCD variations along the WCI are far more correlated with STI winds (-0.73) than with local alongshore winds (-0.25). This now discussed in the revised text (P13 L27-29).

*Page 12, lines 14-21, and Fig. 11b. This is an interesting distillation of results, though I still have reservations of robustness similar and related to what has been noted in comments above. I think it would be useful to find a way to split out the DMI values such that those that pass threshold and can be classified as positive / negative IOD are distinct from those that do not (i.e., normal and IOD states are clearly delineated). The regressions that are shown do not add much insight, but if they are retained then they should be performed only on points that are +/- IOD states and not inclusive of all positive of negative values.*

The updated Figure 12 (previous Fig. 11b) now highlights with different color codes the years corresponding to positive and negative IOD events (red and blue dots respectively) along with neutral years (black dots). Positive (resp. negative) IOD events are identified as years when the DMI index exceeds one standard deviation (resp. is lesser than minus one standard deviation). All other years are classified as normal years. Regression lines have been removed from the revised manuscript, as they did not provide much insight, as pointed out by the reviewer.

*However, even doing this, I am skeptical that particularly insightful result will be obtained. For the data that are shown that do represent + /- IOD occurrence, there is not really an associated systematic deepening / shoaling of OCD. Certainly the negative IOD case has a number of either OCD result. But there is also a positive IOD case with (slightly) shallowing OCD. It also appears that the years with observed and notable anoxic events tend to be during negative DMI (thought not necessarily negative IOD) but there is also a positive DMI case and a case that is almost uniformly normal (i.e., DMI _ 0 and OCD _ 0). Given these results, I do consider that the data are suggestive of positive IOD events influencing appearance of low oxygen in WCI but would be hesitant to make categorical conclusions. Overall, I think the authors need to be more even handed when reporting what is revealed in Fig.11b.*

New figure 13 (old Figure 12) clearly demonstrates that, on an average, positive IOD events lead to a deepening of the TCD/OCD along the west coast of India. Similarly, negative IOD events are associated, in general, with a shoaling of TCD/OCD along the WCI. However, although the correlation between DMI and WCI OCD is significant (0.67), it is not close to one, demonstrating that there are clear departures from this average picture. This is actually what is shown on new Figure 12 (previous Figure 11b), where positive IOD can be related sometimes with insignificant OCD anomalies and negative IOD can even be related in some instances with OCD deepening. This suggests that IOD is not the only process that controls the interannual OCD variations along the WCI. Other processes, such as local wind fluctuations or biological processes can indeed counteract the IOD influence. This discussion has been expanded in the revised text (P14 L14-28) and further discussed in the last section. We now clearly state in the abstract that positive IODs suppress favorable conditions for anoxic events along the WCI and negative or neutral IODs are a necessary but not sufficient condition for anoxia to occur along the WCI (P1 L18-24)

*Page 12, lines 21-27, and Fig. 12. This figure is also quite interesting. It suggests that both IOD phases lead to positive OCD and TCD displacement during SON, although for negative IOD this may not be statistically significant. What I find curious is that taux during SON in STI that has opposite sign between positive and negative IOD. Would this not lead to opposite oceanic response (i.e., upwelling vs. downwelling) between the two IOD phases, which then presumably has contrasting impact on thermocline displacement if this response translates northwards as coastal KW as authors have argued? The authors do note that taux condition for negative IOD is not as "robust" as that for positive IOD, however the relative values of anomaly for October (highest magnitude for SON period) are not strikingly distinct. Which begs the question as to how much OCD / TCD response one can*

*The referee comments are italicized.* Answers in regular typeface. Actions taken in red.

*ascribe to STI wind stress condition.*

First, we believe that there is a misunderstanding here. In this Figure, the negative IOD composite time series has been multiplied by minus one to ease comparison with positive events (as stated in the Figure caption). Hence, positive IOD events are associated with OCD and TCD deepening and negative zonal wind stress anomalies at the STI. In contrast, negative IOD events are associated with OCD/TCD shoaling (although hardly significant) and positive zonal wind stress anomalies at the STI. These evolutions are hence consistent with a Kelvin wave response at the STI to zonal wind variations associated with the IOD fluctuations. To avoid confusion, we now mention explicitly in the main text (P14 L21-22), figure caption and figure label (« -1 x Negative-IOD ») that the the negative IOD composite time series has been multiplied by minus one to ease comparison with positive events. To better asses how much OCD response can be ascribed to STI wind stresss variations, the new Figure 11b compares interannual time series of WCI OCD and STI Taux: these two parameters display a strong and significant correlation (-0.73), demonstrating that more than 50% of the OCD variability along the WCI can be attributed to STI wind variations. This will be more clearly stated in the updated text (e.g. P13 L27-29).

*Page 12, slide 30. Instead of "identified fifteen years ago", reiterating citation of relevant source material would be best.*

This has been done.

*Page 20, line 8. Title for this reference is incorrect.*

This has been corrected.

*Comments on Figures*
*General comment. There are several figures that include isolines superimposed on a second variable where a color scale is applied. In almost all cases, there is a need for additional labeling of the superimposed contour lines on the plot so these distributions can be grasped. There is also a need to note the contour intervals in the associated captions.*

This has been done.

*Figure 4. For panels 4b and 4d, the red contour line demarking thermocline is extremely difficult to see. In addition, the isotherms at depth (DO < 80 microM) are also difficult to see. For panels 4a and 4c, the 15_N line is difficult to spot as well.*

These colors have been changed to white.

*Figure 9. In panel 9a, white color in the IO domain has non-unique meaning. It can be either masked data or the transition from positive to negative correlation value. A way of uniquely distinguishing these should be used. Additionally, the white land mask is also not ideal but does have benefit of land-sea boundary.*

We now use grey shade for the mask.

*Panels 11a and 12 a-c. The combination of blue and black line colors is ineffective. Very difficult to distinguish between these two.*

Black has been changed to red for better visual appeal.

[revised manuscript text omitted]

20  Given the existing high frequency variability, averaging this very limited number of profiles may not provide a value representative of the actual monthly average. As a result, the uneven temporal distribution and the sub-monthly variability do not allow us to build a reliable monthly pluri-annual time series from this dataset. In addition, these observations collected at a single coastal site may be influenced by local processes and hence may not be representative of the dynamics along the entire WCI. We will hence identify the years when severe oxygen deficiency occurs from Table 1 of Naqvi et al. (2009),

25  which is based on the absence of nitrate and nitrite and the presence of hydrogen sulphide in the water column. These years will be further discussed and compared with our model results at the end of this paper.

**2.2 Model description**

The model used in this study couples the NEMO (Nucleus for European Modelling of the Ocean; Madec, 2008) physical ocean component with the PISCES (Pelagic Interaction Scheme for Carbon and Ecosystem Studies; Aumont et al., 2015)

30  biogeochemical component through the OASIS3 (Valcke, 2013) coupler. The PISCES model has 24 compartments, which include two sizes of sinking particles and four "living" biological pools, representing two phytoplankton (nano-phytoplankton and diatoms) and two zooplankton (microzooplankton and meso-zooplankton) size classes. Phytoplankton

I. Suresh 26/12/2016 10:31 AM

I. Suresh 26/12/2016 10:31 AM

I. Suresh 26/12/2016 10:31 AM

I. Suresh 26/12/2016 10:31 AM

I. Suresh 26/12/2016 10:31 AM

I. Suresh 26/12/2016 10:31 AM

I. Suresh 26/12/2016 10:31 AM

I. Suresh 26/12/2016 10:31 AM

I. Suresh 26/12/2016 10:31 AM
Moved down [2]: Fig.

I. Suresh 26/12/2016 10:31 AM

growth is limited by five nutrients: NO3, NH4, PO4, SiO4, and Fe. The ratios among C, N, and P are kept constant for the "living" compartments, at values proposed by Takahashi et al. (1985). The internal Fe contents of both phytoplankton groups and Si contents of diatoms are prognostically simulated as a function of ambient concentrations in nutrients and light level. Details on the red-green-blue model from which light penetration profiles are calculated are given in Lengaigne et al. (2007).

5   The Chl/C ratio is modeled using a modified version of the photo-adaptation model by Geider et al. (1998). Dissolved oxygen is prognostic and evolves in response to physical conditions (advection, mixing), biological sources and sinks and the air-sea fluxes:

$$\partial_t O_2 = \underbrace{\left(\frac{\partial O_2}{\partial t}\right)_{Dyn}}_{Dynamical\ Transport} + \underbrace{\left(\frac{\partial O_2}{\partial t}\right)_{Bio}}_{Biological\ sources\ \&\ sinks} + J_{flux}$$

10   where $(\partial O_2/\partial t)_{Bio}$ includes all biological processes affecting the concentration of $O_2$, $(\partial O_2/\partial t)_{Dyn}$ accounts for large scale and turbulent transport of oxygen and $J_{flux}$ is the contribution of $O_2$ air–sea fluxes. The response of oxygen to biological processes $(\partial O_2/\partial t)_{Bio}$ is computed as follows:

$$\left(\frac{\partial O_2}{\partial t}\right)_{Bio} = \underbrace{(R_{o:c}^1 + R_{o:c}^2)(\mu_{NO_3}^P P + \mu_{NO_3}^D D)}_{New\ Production}$$
$$+ \underbrace{R_{o:c}^1(\mu_{NH_4}^P P + \mu_{NH_4}^D D)}_{Regenerated\ Production} - \underbrace{\lambda_{DOC}^* f(O_2) DOC}_{Reminralization}$$
$$\underbrace{-G^Z Z - G^M M}_{Respiration} - \underbrace{R_{o:c}^2 Nitrif}_{Nitrification}$$

[revised manuscript text omitted]

5    A seasonally shallow OCD combined with a larger interannual variability in fall creates a window of opportunity for the occurrence of coastal anoxic events. Figs. 7d and 7e display the monthly percentages of occurrence of hypoxic profiles from CaTS (on the shelf) data and the model (offshore). While the general patterns of oxygen/oxycline variability on the shelf and offshore to the coast remain similar, the actual upper ocean oxygen content and vertical oxygen profiles are different. Hence, we have used different thresholds to detect hypoxic profiles in the observation and the model. Consistent with previous
10   literature, anoxic events are most likely to occur from August to November in the model and the shelf data, as expected from the very shallow background oxycline at that time of the year. This justifies our focus on the fall period for analyzing the processes that drive the modeled interannual variability of the WCI oxycline in the following.

We previously demonstrated a tight relationship between the seasonal variability of OCD and TCD in the eastern Arabian
15   Sea. Fig. 8a exhibits a similar relation in large portions of the northern Indian Ocean for fall interannual OCD and TCD anomalies in the model. A comparison with observations is unfortunately not feasible due to lack of a basin-scale dataset for interannual OCD anomalies. The correlation between interannual OCD and TCD anomalies are, in general, slightly weaker than that at seasonal timescales (Fig. 5b), but remain high in a large part of the Indian Ocean north of 5°N, generally exceeding 0.8 in the entire Bay of Bengal and in the eastern Arabian Sea, and in particular in the WCI box (~0.95 correlation, Fig. 8b).

20   The influence of remote forcing at WCI is further established in Fig. 9. Due to unavailability of the continuous oxygen observations (see Section 2.1), we cannot directly evaluate the modeled oxygen interannual variability in the WCI region. We however evaluate the modeled TCD interannual variability, which is closely tied to the OCD interannual variability (~0.95 correlation, Fig. 8b). The modeled interannual TCD anomalies in the WCI box (red curve on Fig. 9a) agree well (0.84 correlation) with sea-level interannual anomalies (a good proxy for TCD variations in stratified regions) from altimeter
25   measurements (blue curve on Fig. 9a) during fall. Despite instances, when the model agreement is weaker, like during 2002-2006 period, both the modeled TCD and the observed sea level indicate strongest thermocline shoaling in fall 1999 and deeper than usual thermocline in fall 1994, 1997 and 2008. We will exploit this ability of our model to capture the observed TCD interannual variations along the WCI to further examine the processes responsible for the OCD interannual variability. Along with fall TCD and sea-level anomalies at WCI, Fig. 9a also displays the fall interannual anomalies of zonal winds at
30   the STI (black dashed curve on Fig. 9a; box with dashed frame in Figs. 9b and 9c). The good phase agreement between the interannual fluctuations of zonal winds at STI and both modeled TCD and observed SLA (-0.65 and -0.69 correlation

I. Suresh 26/12/2016 10:31 AM

I. Suresh 26/12/2016 10:31 AM
Deleted: coastal anoxic events to occur during September-November (hereafter SON). This is consistent with previous observations of oxygen deficiency along the WCI (e.g., Naqvi et al., 2000, 2009; Gupta et al., 2016). ... [16]

I. Suresh 26/12/2016 10:31 AM

I. Suresh 26/12/2016 10:31 AM

respectively, Fig. 9a) at WCI is a strong indication that the interannual variations along the WCI are strongly influenced by the wind variations at the STI, through coastal Kelvin waves propagation, as demonstrated by Suresh et al. (2016) for the seasonal timescale. Fig. 9b shows the correlation of interannual anomalies of model TCD everywhere in the northern Indian Ocean with the time series of model TCD in the WCI box during fall. Fig. 9c shows a similar analysis, but for the observed SLA. In both model and observations, the thermocline depth variations along the WCI in fall are associated with basin-scale coherent signals in thermocline depth, sea level and wind stresses. Those basin-scale patterns are very similar to those associated with a positive phase of the IOD mode, with upwelling in the Eastern Equatorial Indian Ocean (EEIO) and a downwelling in the western Indian Ocean. In the following section, we will show how IOD events influence the fall thermocline and oxycline depths off the WCI.

[revised manuscript text omitted]

The influence of IOD is further shown to be larger for positive than its negative phase. Our results suggest that part of the weaker WCI oxycline depth response during negative IOD may be explained by the weaker windstress anomalies at the STI

10 associated with negative IOD events. This weaker wind amplitude could simply be related to the tendency of negative IOD events to be weaker than their positive counterpart (Saji and Yamagata 2003; Hong et al. 2008; Cai et al. 2013) or to asymmetries in the spatial patterns of winds associated associated with the non-linear response of deep atmospheric convection to SST anomalies of each phase of the IOD. A more precise understanding of this asymmetry would require an in-depth investigation of the processes that control the wind variations at the STI and the thermocline along the WCI in

15 response to positive and negative IOD events.

Our findings partly explain the substantial year-to-year changes in both the duration and intensity of the observed seasonal oxygen deficiency over the western Indian shelf (Naqvi et al., 2009). None of the anoxic events reported by Naqvi et al. (2009) (black stars in Fig. 12) lies on the upper right quadrant of the scatterplot shown in Fig. 12, indicating that positive IODs systematically prevent the occurrence of anoxic events. For instance, the relaxation of anoxic condition in early fall

20 1997 reported by Naqvi et al. (2009) is in line with the occurrence of very strong positive IOD during that year. Most anoxic events are found in the lower left quadrant, i.e., near neutral or negative IOD conditions and anomalously shallow offshore oxycline. Neutral or negative IOD years are however not necessarily anoxic, indicating that a neutral or negative IOD is a necessary but not a sufficient condition for severe anoxia. A recent study by Gupta et al. (2016) revealed that the oxygen deficiency in 1959 along the WCI was more severe than in 2012, a conclusion consistent with the occurrence of a negative

25 IOD in 1959 and a positive one in 2012. Similarly, in-situ measurements also revealed that subsurface oxygen concentrations were significantly lower at the turn of the $20^{th}$ century than in the 70's (Naqvi et al., 2009): our simulation exhibit a similar behavior (see Fig. 11a), showing many years with shallower than normal OCD in the later period and systematically deeper than normal OCD during 1970's. The causes for those decadal variations need to be investigated in greater detail.

The ~0.7 correlation between IOD variability and oxycline variations along the WCI implies that ~50% of the interannual

30 oxycline variance is explained by the IOD at this location. This relationship between the IOD and year-to-year variations of seasonal anoxic conditions along the shelf may facilitate advance warning for the possible occurrence of severe anoxic

I. Suresh 26/12/2016 10:31 AM

I. Suresh 26/12/2016 10:31 AM
Deleted: Although our results suggest that part of this asymmetry may be explained by the difference in the strength of winds at STI associated with each phase of the IOD, 
[revised manuscript text omitted]